# Conservative Offline Distributional Reinforcement Learning

**Yecheng Jason Ma, Dinesh Jayaraman, Osbert Bastani**
University of Pennsylvania
{jasonyma, dineshj, obastani}@seas.upenn.edu

## Abstract

Many reinforcement learning (RL) problems in practice are *offline*, learning purely from observational data. A key challenge is how to ensure the learned policy is safe, which requires quantifying the risk associated with different actions. In the online setting, distributional RL algorithms do so by learning the distribution over returns (i.e., cumulative rewards) instead of the expected return; beyond quantifying risk, they have also been shown to learn better representations for planning. We propose Conservative Offline Distributional Actor Critic (CODAC), an offline RL algorithm suitable for both risk-neutral and risk-averse domains. CODAC adapts distributional RL to the offline setting by penalizing the predicted quantiles of the return for out-of-distribution actions. We prove that CODAC learns a conservative return distribution—in particular, for finite MDPs, CODAC converges to an uniform lower bound on the quantiles of the return distribution; our proof relies on a novel analysis of the distributional Bellman operator. In our experiments, on two challenging robot navigation tasks, CODAC successfully learns risk-averse policies using offline data collected purely from risk-neutral agents. Furthermore, CODAC is state-of-the-art on the D4RL MuJoCo benchmark in terms of both expected and risk-sensitive performance. Code is available at: https://github.com/JasonMa2016/CODAC

## 1 Introduction

In many applications of reinforcement learning, actively gathering data through interactions with the environment can be risky and unsafe. Offline (or batch) reinforcement learning (RL) avoids this problem by learning a policy solely from historical data (called *observational data*) [9, 22, 23].

A shortcoming of most existing approaches to offline RL [11, 46, 20, 21, 48, 18] is that they are designed to maximize the expected value of the cumulative reward (which we call the *return*) of the policy. As a consequence, they are unable to quantify risk and ensure that the learned policy acts in a safe way. In the online setting, there has been recent work on *distributional* RL algorithms [7, 6, 27, 38, 17], which instead learn the full distribution over future returns. They can use this distribution to plan in a way that avoids taking risky, unsafe actions. Furthermore, when coupled with deep neural network function approximation, they can learn better state representations due to the richer distributional learning signal [4, 26], enabling them to outperform traditional RL algorithms even on the risk-neutral, expected return objective [4, 7, 6, 47, 14].

We propose Conservative Offline Distributional Actor-Critic (CODAC), which adapts distributional RL to the offline setting. A key challenge in offline RL is accounting for high uncertainty on out-of-distribution (OOD) state-action pairs for which observational data is limited [23, 20]; the value estimates for these state-action pairs are intrinsically high variance, and may be exploited by the policy without correction due to the lack of online data gathering and feedback. We build on conservative $Q$-learning [21], which penalizes $Q$ values for OOD state-action pairs to ensure that

the learned $Q$-function lower bounds the true $Q$-function. Analogously, CODAC uses a penalty to ensure that the quantiles of the learned return distribution lower bound those of the true return distribution. Crucially, the lower bound is data-driven and selectively penalizes the quantile estimates of state-actions that are less frequent in the offline dataset; see Figure 1.

We prove that for finite MDPs, CODAC converges to an estimate of the return distribution whose quantiles uniformly lower bound the quantiles of the true return distribution; in addition, this data-driven lower bound is tight up to the approximation error in estimating the quantiles using finite data. Thus, CODAC obtains a uniform lower bound on *all* integrations of the quantiles, including the standard RL objective of expected return, the risk-sensitive conditional-value-at-risk (CVaR) objective [35], as well as many other risk-sensitive objectives. We additionally prove that CODAC expands the gap in quantile estimates between in-distribution and OOD actions, thus avoiding overconfidence when extrapolating to OOD actions [11].

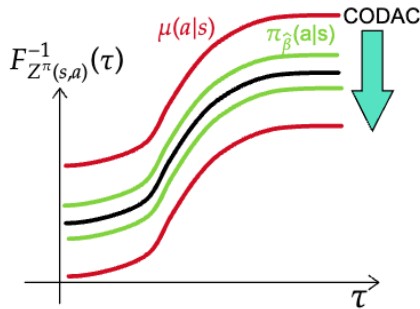

Figure 1: CODAC obtains conservative estimates of the true return quantiles (black); it penalizes out-of-distribution actions, $\mu(a \mid s)$, more heavily than in-distribution actions, $\pi_{\hat{\beta}}(a \mid s)$.

Our theoretical guarantees rely on novel techniques for analyzing the distributional Bellman operator, which is challenging since it acts on the infinite-dimensional function space of return distributions (whereas the traditional Bellman operator acts on a finite-dimensional vector space). We provide several novel results that may be of independent interest; for instance, our techniques can be used to bound the error of the fixed-point of the empirical distributional Bellman operator; see Appendix A.6.

Finally, to obtain a practical algorithm, CODAC builds on existing distributional RL algorithms by integrating conservative return distribution estimation into a quantile-based actor-critic framework.

In our experiments, we demonstrate the effectiveness of CODAC on both risk-sensitive and risk-neutral RL. First, on two novel risk-sensitive robot navigation tasks, we show that CODAC successfully learns risk-averse policies using offline datasets collected purely from a risk-neutral agent, a challenging task that all our baselines fail to solve. Next, on the D4RL Mujoco suite [10], a popular offline RL benchmark, we show that CODAC achieves state-of-art results on both the original risk-neutral version as well a modified risk-sensitive version [43]. Finally, we empirically show that CODAC computes quantile lower-bounds and gap-expanded quantiles even on high-dimensional continuous-control problems, validating our key theoretical insights into the effectiveness of CODAC.

**Related work.** There has been growing interest in offline (or batch) RL [22, 23]. The key challenge in offline RL is to avoid overestimating the value of out-of-distribution (OOD) actions rarely taken in the observational dataset [40, 44, 20]. The problem is that policy learning optimizes against the value estimates; thus, even if the estimation error is i.i.d., policy optimization biases towards taking actions with high variance value estimates, since some of these values will be large by random chance. In risk-sensitive or safety-critical settings, these actions are exactly the ones that should be avoided.

One solution is to constrain the learned policy to take actions similar to the ones in the dataset (similar to imitation learning)—e.g., by performing support matching [46] or distributional matching [20, 12]. However, these approaches tend to perform poorly when data is gathered from suboptimal policies. An alternative is to regularize the $Q$-function estimates to be conservative at OOD actions [21, 48, 18]. CODAC builds on these approaches, but obtains conservative estimates of all quantile values of the return distribution rather than just the expected return. Traditionally, the literature on off-policy evaluation (OPE) [32, 16, 39, 25, 37] aims to estimate the expected return of a policy using pre-collected offline data; CODAC proposes an OPE procedure amenable to all objectives that can be expressed as integrals of the return quantiles. Consequently, our fine-grained approach not only enables risk-sensitive policy learning, but also improves performance on risk-neutral domains.

In particular, CODAC builds on recent works on distributional RL [4, 7, 6, 47], which parameterize and estimate the entire return distribution instead of just a point estimate of the expected return (i.e., the $Q$-function) [29, 28]. Distributional RL algorithms have been shown to achieve state-of-art performance on Atari and continuous control domains [14, 3]; intuitively, they provide richer training signals that stabilize value network training [4]. Existing distributional RL algorithms parameterize

the policy return distribution in many different ways, including canonical return atoms [4], the expectiles [36], the moments [31], and the quantiles [7, 6, 47]. CODAC builds on the quantile approach due to its suitability for risk-sensitive policy optimization. The quantile representation provides an unified framework for optimizing different objectives of interest [7], such as the risk-neutral expected return, and a family of risk-sensitive objectives representable by the quantiles; this family includes, for example, the conditional-value-at-risk (CVaR) return [35, 5, 38, 17], the Wang measure [45], and the cumulative probability weighting (CPW) metric [42]. Recent work has provided theoretical guarantees on learning CVaR policies [17]; however, their approach cannot provide bounds on the quantiles of the estimated return distribution, which is significantly more challenging since there is no closed-form expression for the Bellman update on the return quantiles.

Finally, there has been some recent work adapting distributional RL to the offline setting [1, 43]. First, REM [1] builds on QR-DQN [7], an online distributional RL algorithm; however, REM can be applied to regular DQN [29] and does not directly utilize the distributional aspect of QR-DQN. The closest work to ours is ORAAC [43], which uses distributional RL to learn a CVaR policy in the offline setting. ORAAC uses imitation learning to avoid OOD actions and stay close to the data distribution. As discussed above, imitation learning strategies can perform poorly unless the dataset comes from an optimal policy; in our experiments, we find that CODAC significantly outperforms ORAAC. Furthermore, unlike ORAAC, we provide theoretical guarantees on our approach.

## 2 Background

**Offline RL.** Consider a Markov Decision Process (MDP) [33] $(\mathcal{S}, \mathcal{A}, P, R, \gamma)$, where $\mathcal{S}$ is the state space, $\mathcal{A}$ is the action space, $P(s' \mid s, a)$ is the transition distribution, $R(r \mid s, a)$ is the reward distribution, and $\gamma \in (0, 1)$ is the discount factor, and consider a stochastic policy $\pi(a \mid s) : \mathcal{S} \to \Delta(\mathcal{A})$. A *rollout* using $\pi$ from state $s$ using initial action $a$ is the random sequence $\xi = ((s_0, a_0, r_0), (s_1, a_1, r_1), ...)$ such that $s_0 = s$, $a_0 = a$, $a_t \sim \pi(\cdot \mid s_t)$ (for $t > 0$), $r_t \sim R(\cdot \mid s_t, a_t)$, and $s_{t+1} \sim P(\cdot \mid s_t, a_t)$; we denote the distribution over rollouts by $D^\pi(\xi \mid s, a)$. The $Q$-function $Q^\pi : \mathcal{S} \times \mathcal{A} \to \mathbb{R}$ of $\pi$ is its expected discounted cumulative return $Q^\pi(s, a) = \mathbb{E}_{D^\pi(\xi \mid s, a)}[\sum_{t=0}^\infty \gamma^t r_t]$. Assuming the rewards satisfy $r_t \in [R_{\min}, R_{\max}]$, then we have $Q^\pi(s, a) \in [V_{\min}, V_{\max}] \subseteq [R_{\min}/(1 - \gamma), R_{\max}/(1 - \gamma)]$.

A standard goal of reinforcement learning (RL), which we call *risk-neutral* RL, is to learn the optimal policy $\pi^*$ such that $Q^{\pi^*}(s, a) \geq Q^\pi(s, a)$ for all $s \in \mathcal{S}, a \in \mathcal{A}$ and all $\pi$.

In offline RL, the learning algorithm only has access to a fixed dataset $\mathcal{D} := \{(s, a, r, s')\}$, where $r \sim R(\cdot \mid s, a)$ and $s' \sim P(\cdot \mid s, a)$. The goal is to learn the optimal policy without any interaction with the environment. Though we do not assume that $\mathcal{D}$ necessarily comes from a single behavior policy, we define the empirical behavior policy to be $\hat{\pi}_\beta(a \mid s) := \frac{\sum_{s', a' \in \mathcal{D}} \mathbb{1}(s'=s, a'=a)}{\sum_{s' \in \mathcal{D}} \mathbb{1}(s'=s)}$. With slight abuse of notation, we write $(s, a, r, s') \sim \mathcal{D}$ to denote a uniformly random sample from the dataset. Also, in this paper, we broadly refer to actions not drawn from $\hat{\pi}_\beta(\cdot \mid s)$ (i.e., low probability density) as out-of-distribution (OOD).

Fitted $Q$-evaluation (FQE) [9, 34] uses $Q$-learning for offline RL, which leverages the fact that $Q^\pi = \mathcal{T}^\pi Q^\pi$ is the unique fixed point of the Bellman operator $\mathcal{T}^\pi : \mathbb{R}^{|\mathcal{S}||\mathcal{A}|} \to \mathbb{R}^{|\mathcal{S}||\mathcal{A}|}$ defined by

$$\mathcal{T}^\pi Q(s, a) = \mathbb{E}_{R(r \mid s, a)}[r] + \gamma \cdot \mathbb{E}_{P^\pi(s', a' \mid s, a)}[Q(s', a')],$$

where $P^\pi(s', a' \mid s, a) = P(s' \mid s, a)\pi(a' \mid s')$. In the offline setting, we do not have access to $\mathcal{T}^\pi$; instead, FQE uses an approximation $\hat{\mathcal{T}}^\pi$ obtained by replacing $R$ and $P$ in $\mathcal{T}^\pi$ with estimates $\hat{R}$ and $\hat{P}$ based on $\mathcal{D}$. Then, we can estimate $Q^\pi$ by starting from an arbitrary $\hat{Q}^0$ and iteratively computing

$$\hat{Q}^{k+1} := \arg\min_Q \mathcal{L}(\hat{Q}, \hat{\mathcal{T}}^\pi \hat{Q}^k) \quad \text{where} \quad \mathcal{L}(Q, Q') = \mathbb{E}_{\mathcal{D}(s, a)}\left[(Q(s, a) - Q'(s, a))^2\right].$$

Assuming we search over the space of all possible $Q$ (i.e., do not use function approximation), then the minimizer is $\hat{Q}^{k+1} = \hat{\mathcal{T}}^\pi \hat{Q}^k$, so $\hat{Q}^k = (\hat{\mathcal{T}}^\pi)^k Q^0$. If $\hat{\mathcal{T}}^\pi = \mathcal{T}^\pi$, then $\lim_{k \to \infty} \hat{Q}^k = Q^\pi$.

**Distributional RL.** In distributional RL, the goal is to learn the distribution of the discounted cumulative rewards (i.e., *returns*) [4]. Given a policy $\pi$, we denote its return distribution as the random variable $Z^\pi(s, a) = \sum_{t=0}^\infty \gamma^t r_t$, which is a function of a random rollout $\xi \sim D^\pi(\cdot \mid s, a)$;

note that $Z^\pi$ includes three sources of randomness: (1) the reward $R(\cdot \mid s, a)$, (2) the transition $P(\cdot \mid s, a)$, and (3) the policy $\pi(\cdot \mid s)$. Also, note that $Q^\pi(s, a) = \mathbb{E}_{D^\pi(\xi \mid s, a)}[Z^\pi(s, a)]$. Analogous to $Q$-function Bellman operator, the distributional Bellman operator for $\pi$ is

$$\mathcal{T}^\pi Z(s, a) \overset{D}{:=} r + \gamma Z(s', a') \quad \text{where} \quad r \sim R(\cdot \mid s, a), \ s' \sim P(\cdot \mid s, a), \ a' \sim \pi(\cdot \mid s'), \quad (1)$$

where $\overset{D}{=}$ indicates equality in distribution. As with $Q^\pi$, $Z^\pi$ is the unique fixed point of $\mathcal{T}^\pi$ in Eq. 1.

Next, let $F_{Z(s,a)}(x) : [V_{\min}, V_{\max}] \to [0, 1]$ be the cumulative density function (CDF) for return distribution $Z(s, a)$, and $F_{R(s,a)}$ be the CDF of $R(\cdot \mid s, a)$ Then, we have the following equality, which captures how the distributional Bellman operator $\mathcal{T}^\pi$ operates on the CDF $F_{Z(s,a)}$ [17]:

$$F_{\mathcal{T}^\pi Z(s,a)}(x) = \sum_{s',a'} P^\pi(s', a' \mid s, a) \int F_{Z(s',a')} \left( \frac{x - r}{\gamma} \right) dF_{R(s,a)}(r). \quad (2)$$

Let $X$ and $Y$ be two random variables. Then, the *quantile function* (i.e., inverse CDF) $F_X^{-1}$ of $X$ is $F_X^{-1}(\tau) := \inf\{x \in \mathbb{R} \mid \tau \le F_X(x)\}$, and the *p-Wasserstein distance* between $X$ and $Y$ is $W_p(X, Y) = (\int_0^1 |F_Y^{-1}(\tau) - F_X^{-1}(\tau)|^p d\tau)^{1/p}$. Then, the distributional Bellman operator $\mathcal{T}^\pi$ is a $\gamma$-contraction in the $W_p$ [4]—i.e., letting $\bar{d}_p(Z_1, Z_2) := \sup_{s,a} W_p(Z_1(s, a), Z_2(s, a))$ be the largest Wasserstein distance over $(s, a)$, and $\mathcal{Z} = \{Z : \mathcal{S} \times \mathcal{A} \to \mathcal{P}(\mathbb{R}) \mid \forall(s, a) . \mathbb{E}[|Z(s, a)|^p] < \infty\}$ be the space of distributions over $\mathbb{R}$ with bounded $p$-th moment, then

$$\bar{d}_p(\mathcal{T}^\pi Z_1, \mathcal{T}^\pi Z_2) \le \gamma \bar{d}_p(Z_1, Z_2) \qquad (\forall Z_1, Z_2 \in \mathcal{Z}). \quad (3)$$

As a result, $Z^\pi$ may be obtained by iteratively applying $\mathcal{T}^\pi$ to an initial distribution $Z$.

As before, in the offline setting, we can approximate $\mathcal{T}^\pi$ by $\hat{\mathcal{T}}^\pi$ using $\mathcal{D}$. Then, we can compute $Z^\pi$ (represented as $F_{Z(s,a)}^{-1}$; see below) by starting from an arbitrary $\hat{Z}^0$, and iteratively computing

$$\hat{Z}^{k+1} = \arg\min_Z \mathcal{L}_p(Z, \hat{\mathcal{T}}^\pi \hat{Z}^k) \quad \text{where} \quad \mathcal{L}_p(Z, Z') = \mathbb{E}_{\mathcal{D}(s,a)} \left[ W_p(Z(s, a), Z'(s, a))^p \right]. \quad (4)$$

We call this procedure *fitted distributional evaluation (FDE)*.

One distributional RL algorithmic framework is quantile-based distributional RL [7, 6, 47, 27, 38, 43], where the return distribution $Z$ is represented by its quantile function $F_{Z(s,a)}^{-1}(\tau) : [0, 1] \to \mathbb{R}$. Given a distribution $g(\tau)$ over $[0, 1]$, the *distorted expectation* of $Z$ is $\Phi_g(Z(s, a)) = \int_0^1 F_{Z(s,a)}^{-1}(\tau) g(\tau) d\tau$, and the corresponding policy is $\pi_g(s) := \arg\max_a \Phi_g(Z(s, a))$ [7]. If $g = \text{Uniform}([0, 1])$, then $Q^\pi(s, a) = \Phi_g(Z(s, a))$; alternatively, $g = \text{Uniform}([0, \xi])$ corresponds to the CVaR [35, 5, 6] objective, where only the bottom $\xi$-percentile of the return is considered. Additional risk-sensitive objectives are also compatible. For example, CPW [42] amounts to $g(\tau) = \tau^\beta / (\tau^\beta + (1 - \tau)^\beta)^{\frac{1}{\beta}}$, and Wang [45] has $g(\tau) = F_\mathcal{N}(F_\mathcal{N}^{-1}(\tau) + \beta)$, where $F_\mathcal{N}$ is the standard Gaussian CDF.

A drawback of FDE is that it does not account for estimation error, especially for pairs $(s, a)$ that rarely appear in the given dataset $\mathcal{D}$; thus, $\hat{Z}^k(s, a)$ may be an overestimate of $Z^k(s, a)$ [12, 20, 21], even in distributional RL (since the learned distribution does not include randomness in the dataset) [14, 3]. Importantly, since we act by optimizing with respect to $\hat{Z}^k(s, a)$, the optimization algorithm will exploit these errors, biasing towards actions with higher uncertainty, which is the opposite of what is desired. In Section 3, we propose and analyze a penalty designed to avoid this issue.

## 3 Conservative offline distributional policy evaluation

We describe our algorithm for computing a conservative estimate of $Z^\pi(s, a)$, and provide theoretical guarantees for finite MDPs. In particular, we modify Eq. 4 to include a penalty term:

$$\tilde{Z}^{k+1} = \arg\min_Z \alpha \cdot \mathbb{E}_{U(\tau), \mathcal{D}(s,a)} \left[ c_0(s, a) \cdot F_{Z(s,a)}^{-1}(\tau) \right] + \mathcal{L}_p(Z, \hat{\mathcal{T}}^\pi \tilde{Z}^k) \quad (5)$$

for some state-action dependent scale factor $c_0$; here, $U = \text{Uniform}([0, 1])$. This objective adapts the conservative penalty in prior work [21] to the distributional RL setting; in particular, the first term in

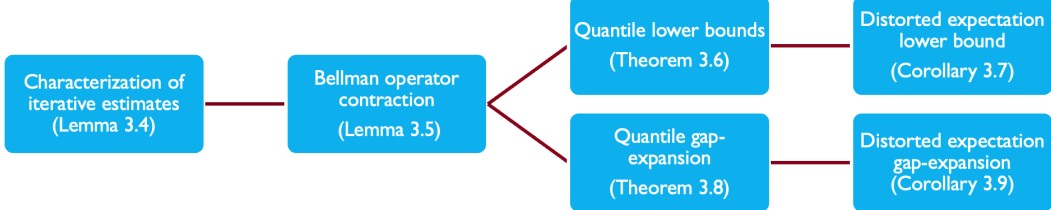

Figure 2: Overview of our theoretical results.

the objective is a penalty that aims to shrink the quantile values for out-of-distribution (OOD) actions compared to those of in-distribution actions; intuitively, $c_0(s, a)$ should be larger for OOD actions. For now, we let $c_0$ be arbitrary; we describe our choice in Section 4. $\alpha \in \mathbb{R}_{>0}$ is a hyperparameter controlling the magnitude of the penalty term with respect to the usual FDE objective. We call this iterative algorithm *conservative distribution evaluation (CDE)*.

Next, we analyze theoretical properties of CDE in the setting of finite MDPs; Figure 2 overviews these results. First, we prove that CDE iteratively obtains conservative quantile estimates (Lemma 3.4) and defines a contraction operator on return distributions (Lemma 3.5). Then, our main result (Theorem 3.6) is that CDE converges to a fixed point $\tilde{Z}^\pi$ whose quantile function lower bounds that of the true returns $Z^\pi$. We also prove that CDE is *gap-expanding* (Theorem 3.8)—i.e., it is more conservative for actions that are rare in $\mathcal{D}$. These results translate to RL objectives computed by integrating the return quantiles, including expected and CVaR returns (Corollaries 3.7 & 3.9).

We begin by describing our assumptions on the MDP and dataset. First, we assume that our dataset $\mathcal{D}$ has nonzero coverage of all actions for states in the dataset [23, 21].

**Assumption 3.1.** For all $s \in \mathcal{D}$ and $a \in \mathcal{A}$, we have $\hat{\pi}_\beta(a \mid s) > 0$.

This assumption is only needed by our theoretical analysis to avoid division-by-zero and ensure that all estimates are well-defined; alternatively, we could assign a very low value $\hat{\pi}_\beta(a \mid s) := \epsilon$ for all actions not visited at state $s$ in the offline dataset and renormalize $\hat{\pi}_\beta(a \mid s)$ accordingly. Next, we impose regularity conditions on the fixed point $Z^\pi$ of the distributional Bellman operator $\mathcal{T}^\pi$.

**Assumption 3.2.** For all $s \in \mathcal{S}$ and $a \in \mathcal{A}$, $F_{Z^\pi(s,a)}$ is smooth. Furthermore, there exists $\zeta \in \mathbb{R}_{>0}$ such that for all $s \in \mathcal{S}$ and $a \in \mathcal{A}$, $F_{Z^\pi(s,a)}$ is $\zeta$-strongly monotone—i.e., we have $F'_{Z^\pi(s,a)}(x) \geq \zeta$.

The smoothness assumption ensures that the $p$th moments of $Z^\pi(s, a)$ are bounded (since $Z^\pi \in [V_{\min}, V_{\max}]$ is also bounded), which in turn ensures that $Z^\pi \in \mathcal{Z}$. The monotonicity assumption is needed to ensure convergence of $F_{Z^\pi(s,a)}^{-1}$. Next, we assume that the search space in (5) includes all possible functions (i.e., no function approximation).

**Assumption 3.3.** The search space of the minimum over $Z$ in (5) is over all smooth functions $F_{Z(s,a)}$ (for all $s \in \mathcal{S}$ and $a \in \mathcal{A}$) with support on $[V_{\min}, V_{\max}]$.

This assumption is required for us to analytically characterize the solution $\tilde{Z}^{k+1}$ of the CDE objective. Finally, we also assume $p > 1$ (i.e., we use the $p$-Wasserstein distance for some $p > 1$).

Now, we describe our key results. Our first result characterizes the CDE iterates $\tilde{Z}^{k+1}$; importantly, if $c_0(s, a) > 0$, then these iterates encode successively more conservative quantile estimates.

**Lemma 3.4.** *For all $s \in \mathcal{D}$, $a \in \mathcal{A}$, $k \in \mathbb{N}$, and $\tau \in [0, 1]$, we have*

$$F_{\tilde{Z}^{k+1}(s,a)}^{-1}(\tau) = F_{\hat{\mathcal{T}}^\pi \tilde{Z}^k(s,a)}^{-1}(\tau) - c(s,a) \ \ where \ \ c(s,a) = |\alpha p^{-1} c_0(s,a)|^{1/(p-1)} \cdot \mathrm{sign}(c_0(s,a)).$$

We give a proof in Appendix A.1; roughly speaking, it follows by setting the gradient of (5) equal to zero, relying on results from the calculus of variations to handle the fact that $F_{Z(s,a)}^{-1}$ is a function.

Next, we define the *CDE operator* $\tilde{\mathcal{T}}^\pi = \mathcal{O}_c \hat{\mathcal{T}}^\pi$ to be the composition of $\hat{\mathcal{T}}^\pi$ with the *shift operator* $\mathcal{O}_c : \mathcal{Z} \to \mathcal{Z}$ defined by $F_{\mathcal{O}_c Z(s,a)}^{-1}(\tau) = F_{Z(s,a)}^{-1}(\tau) - c(s,a)$; thus, Lemma 3.4 says $\tilde{Z}^{k+1} = \tilde{\mathcal{T}}^\pi \tilde{Z}^k$. Now, we show that $\tilde{\mathcal{T}}^\pi$ is a contraction in the maximum Wasserstein distance $\bar{d}_p$.

**Lemma 3.5.** $\tilde{\mathcal{T}}^\pi$ *is a $\gamma$-contraction in $\bar{d}_p$, so $\tilde{Z}^k$ converges to a unique fixed point $\tilde{Z}^\pi$.*

The first part follows since $\hat{\mathcal{T}}^\pi$ is a $\gamma$-contraction in $\bar{d}_p$ [4, 7], and $\mathcal{O}_c$ is a non-expansion in $\bar{d}_p$, so by composition, $\tilde{\mathcal{T}}^\pi$ is a $\gamma$-contraction in $\bar{d}_p$; the second follows by the Banach fixed point theorem.

Now, our first main theorem says that the fixed point $\tilde{Z}^\pi$ of $\tilde{\mathcal{T}}^\pi$ is a conservative estimate of $Z^\pi$ at all quantiles $\tau$—i.e., CDE computes quantile estimates that lower bound the quantiles of the true return; furthermore, it says that this lower bound is tight.

**Theorem 3.6.** *For any $\delta \in \mathbb{R}_{>0}$, $c_0(s, a) > 0$, with probability at least $1 - \delta$,*

$$F_{Z^\pi(s,a)}^{-1}(\tau) \geq F_{\tilde{Z}^\pi(s,a)}^{-1}(\tau) + (1-\gamma)^{-1} \min_{s',a'}\{c(s', a') - \Delta(s', a')\},$$

$$F_{Z^\pi(s,a)}^{-1}(\tau) \leq F_{\tilde{Z}^\pi(s,a)}^{-1}(\tau) + (1-\gamma)^{-1} \max_{s',a'}\{c(s', a') - \Delta(s', a')\}$$

*for all $s \in \mathcal{D}$, $a \in \mathcal{A}$, and $\tau \in [0, 1]$, where $\Delta(s, a) = \frac{1}{\zeta}\sqrt{\frac{5|\mathcal{S}|}{n(s,a)} \log \frac{4|\mathcal{S}||\mathcal{A}|}{\delta}}$. Furthermore, for $\alpha$ sufficiently large (i.e., $\alpha \geq \max_{s,a}\{\frac{p \cdot \Delta(s,a)^{p-1}}{c_0(s,a)}\}$), we have $F_{Z^\pi(s,a)}^{-1}(\tau) \geq F_{\tilde{Z}^\pi(s,a)}^{-1}(\tau)$.*

We give a proof in Appendix A.2. The first inequality says that the quantile estimates computed by CDE form a lower bound on the true quantiles; this bound is not vacuous as long as $\alpha$ satisfies the given condition. Furthermore, the second inequality states that this lower bound is tight.

Many RL objectives (e.g., expected or CVaR return) are distorted expectations (i.e, integrals of the return quantiles). We can extend Theorem 3.6 to obtain conservative estimates for all such objectives:

**Corollary 3.7.** *For any $\delta \in \mathbb{R}_{>0}$, $c_0(s, a) > 0$, $\alpha$ sufficiently large, and $g(\tau)$, with probability at least $1 - \delta$, for all $s \in \mathcal{D}$, $a \in \mathcal{A}$, we have $\Phi_g(Z^\pi(s, a)) \geq \Phi_g(\tilde{Z}^\pi(s, a))$.*

Choosing $g = \text{Uniform}([0, 1])$ gives $Q^\pi(s, a) \geq \tilde{Q}^\pi(s, a)$—i.e., a lower bound on the $Q$-function. CQL [21] obtains a similar lower-bound; thus, CDE generalizes CQL to other objectives—e.g., it can be used in conjunction with any distorted expectation objective (e.g., CVaR, Wang, CPW, etc.) for risk-sensitive offline RL.

Note that Theorem 3.6 does not preclude the possibility that the lower bounds are more conservative for good actions (i.e., ones for which $\hat{\pi}_\beta(a \mid s)$ is larger). We prove that under the choice[1]

$$c_0(s, a) = \frac{\mu(a \mid s) - \hat{\pi}_\beta(a \mid s)}{\hat{\pi}_\beta(a \mid s)} \tag{6}$$

for some $\mu(a \mid s) \neq \hat{\pi}_\beta(a \mid s)$, then $\tilde{\mathcal{T}}^\pi$ is *gap-expanding*—i.e., the difference in quantile values between in-distribution and out-of-distribution actions is larger under $\tilde{\mathcal{T}}^\pi$ than under $\mathcal{T}^\pi$. Intuitively, $c_0(s, a)$ is large for actions $a$ with higher probability under $\mu$ than under $\hat{\pi}_\beta$ (i.e., an OOD action).

**Theorem 3.8.** *For $p = 2$, $\alpha$ sufficiently large, and $c_0$ as in (6), for all $s \in \mathcal{S}$ and $\tau \in [0, 1]$,*

$$\mathbb{E}_{\hat{\pi}_\beta(a|s)} F_{\tilde{Z}^\pi(s,a)}^{-1}(\tau) - \mathbb{E}_{\mu(a|s)} F_{\tilde{Z}^\pi(s,a)}^{-1}(\tau) \geq \mathbb{E}_{\hat{\pi}_\beta(a|s)} F_{Z^\pi(s,a)}^{-1}(\tau) - \mathbb{E}_{\mu(a|s)} F_{Z^\pi(s,a)}^{-1}(\tau).$$

As before, the gap-expansion property implies gap-expansion of integrals of the quantiles—i.e.:

**Corollary 3.9.** *For $p = 2$, $\alpha$ sufficiently large, $c_0$ as in (6), and any $g(\tau)$, for all $s \in \mathcal{S}$,*

$$\mathbb{E}_{\hat{\pi}_\beta(a|s)} \Phi_g(\tilde{Z}^\pi(s, a)) - \mathbb{E}_{\mu(a|s)} \Phi_g(\tilde{Z}^\pi(s, a)) \geq \mathbb{E}_{\hat{\pi}_\beta(a|s)} \Phi_g(Z^\pi(s, a)) - \mathbb{E}_{\mu(a|s)} \Phi_g(Z^\pi(s, a)).$$

Together, Corollaries 3.7 & 3.9 say that CDE provides conservative lower bounds on the return quantiles while being less conservative for in-distribution actions.

Finally, we briefly discuss the condition on $\alpha$ in Theorems 3.6 & 3.8. In general, $\alpha$ can be taken to be small as long as $\Delta(s, a)$ is small for all $s \in \mathcal{S}$ and $a \in \mathcal{A}$, which in turn holds as long as $n(s, a)$ is large—i.e., the dataset $\mathcal{D}$ has wide coverage.

---

[1] We may have $c_0(s, a) \leq 0$; we can use $c_0'(s, a) = c_0(s, a) + (1 - \min_{s,a} c_0(s, a))$ to avoid this issue.

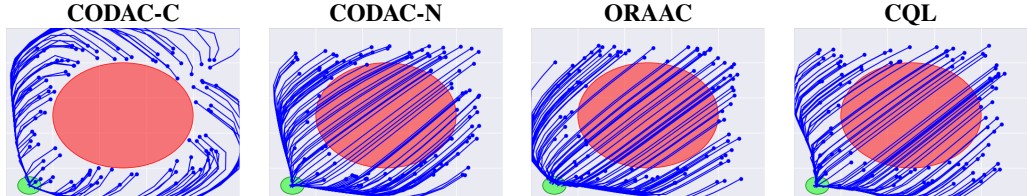

Figure 3: 2D visualization of evaluation trajectories on the Risky PointMass environment. The red region is risky, the solid blue circles indicate initial states, and the blue lines are trajectories. CODAC-C is the only algorithm that successfully avoids the risky region.

## 4   Conservative offline distributional actor critic

Next, we incorporate the distributional evaluation algorithm in Section 3 into an actor-critic framework. Following [21], we propose a min-max objective where the inner loop chooses the current policy to maximize the CDE objective, and the outer loop minimizes the CDE objective for this policy:

$$\hat{Z}^{k+1} = \arg\min_{Z} \max_{\mu} \left\{ \alpha \cdot \mathbb{E}_{U(\tau)} \left[ \mathbb{E}_{\mathcal{D}(s),\mu(a|s)} F^{-1}_{Z(s,a)}(\tau) - \mathbb{E}_{\mathcal{D}(s,a)} F^{-1}_{Z(s,a)}(\tau) \right] + \mathcal{L}_p(Z, \hat{\mathcal{T}}^{\pi^k} \hat{Z}^k) \right\}$$

where we have used $c_0$ as in (6). We can interpret $\mu$ as an actor policy, the first term as the objective for $\mu$, and the overall objective as an actor-critic algorithm [8]. In this framework, a natural choice for $\mu$ is a maximum entropy policy $\mu(a \mid s) \propto \exp(Q(s,a))$ [49]. Then, our objective becomes

$$\hat{Z}^{k+1} = \arg\min_{Z} \left\{ \alpha \cdot \mathbb{E}_{U(\tau)} \left[ \mathbb{E}_{\mathcal{D}(s)} \log \sum_{a} \exp(F^{-1}_{Z(s,a)}(\tau)) - \mathbb{E}_{\mathcal{D}(s,a)} F^{-1}_{Z(s,a)}(\tau) \right] + \mathcal{L}_p(Z, \hat{\mathcal{T}}^{\pi^k} \hat{Z}^k) \right\},$$

where $U = \text{Uniform}([0, 1])$; we provide a derivation in Appendix B. We call this strategy *conservative offline distributional actor critic (CODAC)*. To optimize over $Z$, we represent the quantile function as a DNN $G_\theta(\tau; s, a) \approx F^{-1}_{Z(s,a)}(\tau)$. The main challenge is optimizing the term $\mathcal{L}_p(Z, \hat{\mathcal{T}}^\pi \hat{Z}^k) = W_p(Z, \hat{\mathcal{T}}^\pi \hat{Z}^k)^p$. We do so using distributional temporal-differences (TD) [6]. For a sample $(s, a, r, s') \sim \mathcal{D}$ and $a' \sim \pi(\cdot \mid s')$ and random quantiles $\tau, \tau' \sim U$, we have

$$\mathcal{L}_p(Z, \hat{\mathcal{T}}^\pi \hat{Z}^k) \approx \mathcal{L}_\kappa(\delta; \tau) \qquad \text{where} \qquad \delta = r + \gamma G_{\theta'}(\tau'; s', a') - G_\theta(\tau; s, a).$$

Here, $\delta$ is the distributional TD error, $\theta'$ are the parameters of the target $Q$-network [30], and

$$\mathcal{L}_\kappa(\delta; \tau) = \begin{cases} |\tau - \mathbb{1}(\delta < 0)| \cdot \delta^2/(2\kappa) & \text{if } |\delta| \le \kappa \\ |\tau - \mathbb{1}(\delta < 0)| \cdot (|\delta| - \kappa/2) & \text{otherwise .} \end{cases} \tag{7}$$

is the $\tau$-Huber quantile regression loss at threshold $\kappa$ [15]; then, $\mathbb{E}_{U(\tau)}\mathcal{L}_\kappa(\delta; \tau)$ is an unbiased estimator of the Wasserstein distance that can be optimized using stochastic gradient descent (SGD) [19]. With this strategy, our overall objective can be optimized using any off-policy actor-critic method [24, 13, 11]; we use distributional soft actor-critic (DSAC) [27], which replaces the $Q$-network in SAC [13] with a quantile distributional critic network [7]. We provide the full CODAC pseudocode in Algorithm 1 of Appendix B.

## 5   Experiments

We show that CODAC achieves state-of-the-art results on both risk-sensitive (Sections 5.1 & 5.2) and risk-neutral (Section 5.3) offline RL tasks, including our risky robot navigation and D4RL[2] [10]. We also show that our lower bound (Theorem 3.6) and gap-expansion (Theorem 3.8) results approximately hold in practice, (Section 5.4), validating our theory on CODAC's effectiveness. We provide additional details (e.g., environment descriptions, hyperparameters, and additional results) in Appendix C.

### 5.1   Risky robot navigation

---

[2]The D4RL dataset license is Apache License 2.0.

Table 1: **Risky robot navigation quantitative evaluation**. CODAC-C achieves the best performance on most metrics and is the only method that learns risk-averse behavior. This table is reproduced with standard deviations in Table 6 in Appendix C.

| Algorithm | Risky PointMass | | | | Risky Ant | | | |
|---|---|---|---|---|---|---|---|---|
| | Mean | Median | $CVaR_{0.1}$ | Violations | Mean | Median | $CVaR_{0.1}$ | Violations |
| DSAC (Online) | -7.69 | -3.82 | -49.9 | 94 | -866.1 | -833.3 | -1422.7 | 2247 |
| CODAC-C (Ours) | **-6.05** | -4.89 | **-14.73** | **0.0** | -456.0 | -433.4 | **-686.6** | **347.8** |
| CODAC-N (Ours) | -8.60 | **-4.05** | -51.96 | 108.3 | **-432.7** | **-395.1** | -847.1 | 936.0 |
| ORAAC | -10.67 | -4.55 | -64.12 | 138.7 | -788.1 | -795.3 | -1247.2 | 1196 |
| CQL | -7.51 | -4.18 | -43.44 | 93.4 | -967.8 | -858.5 | -1887.3 | 1854.3 |

**Tasks.** We consider an Ant robot whose goal is to navigate from a random initial state to the green circle as quickly as possible (see Figure 4 for a visualization). Passing through the red circle triggers a high cost with small probability, introducing risk. A risk-neutral agent may pass through the red region, but a risk-aware agent should not. We also consider a PointMass variant for illustrative purposes.

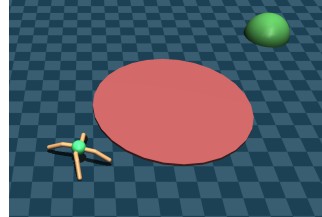

We construct an offline dataset that is the replay buffer of a risk-neutral distributional SAC [27] agent. Intuitively, this choice matches the practical goals of offline RL, where data is gathered from a diverse range of sources with no assumptions on their quality

Figure 4: Risky Ant.

and risk tolerance levels. [23] See Appendix C.1 for details on the environments and datasets.

**Approaches.** We consider two variants of CODAC: (i) CODAC-N, which maximizes the expected return and (ii) CODAC-C, which optimizes the $CVaR_{0.1}$ objective. We compare to Offline Risk-Averse Actor Critic (ORAAC) [43], a state-of-the-art offline risk-averse RL algorithm that combines a distributional critic with an imitation-learning based policy to optimize a risk-sensitive objective, and to Conservative Q-Learning (CQL) [21], a state-of-art offline RL algorithm that is non-distributional.

**Results.** We evaluate each approach using 100 test episodes, reporting the mean, median, and $CVaR_{0.1}$ (i.e., average over bottom 10 episodes) returns, and the total number of violations (i.e., time steps spent inside the risky region), all averaged over 5 random seeds. We also report the performance of the online DSAC agent used to collect data. See Appendix C for details. Results are in Table 1.

**Performance of CODAC.** CODAC-C consistently outperforms the other approaches on the $CVaR_{0.1}$ return, as well as the number of violations, demonstrating that CODAC-C is able to avoid risky actions. It is also competitive in terms of mean return due to its high $CVaR_{0.1}$ performance, but performs slightly worse on median return, since it is not designed to optimize this objective. Remarkably, on Risky PointMass, CODAC-C learns a safe policy that completely avoids the risky region (i.e., zero violations), even though such behavior is absent in the dataset. In Appendix C.1, we also show that CODAC can successfully optimize alternative risk-sensitive objectives such as Wang and CPW.

**Comparison to ORAAC.** While ORAAC also optimizes the CVaR objective, it uses imitation learning to regularize the learned policy to stay close to the empirical behavior policy. However, the dataset contains many sub-optimal trajectories generated early in training, and is furthermore risk-neutral. Thus, imitating the behavioral policy encourages poor performance. In practice, a key use of offline RL is to leverage large datasets available for training, and such datasets will rarely consist of data from a single, high-quality behavioral policy. Our results demonstrate that CODAC is significantly better suited to learned in these settings compared to ORAAC.

**Comparison to CQL.** On Risky PointMass, CQL learns a risky policy with poor tail-performance, indicated by its high median performance but low $CVaR_{0.1}$ performance. Interestingly, its mean performance is also poor; intuitively, the mean is highly sensitive to outliers that may not be present in the training dataset. On Risky Ant, possibly due to the added challenge of high-dimensionality, CQL performs poorly on all metrics, failing to reach the goal and to avoid the risky region. As expected, these results show that accounting for risk is necessary in risky environments.

**Qualitative analysis.** In Figure 3, we show the 100 evaluation rollouts from each policy on Risky PointMass. As can be seen, CODAC-C extrapolates a safe policy that distances itself from the risky

Table 2: **D4RL results.** CODAC achieves the best overall performance in both risk-sensitive (**Left**) and risk-neutral (**Right**) variants of the benchmark. These tables are reproduced with standard deviations in Tables 7 & 9 in Appendix C.

| | Algorithm | Medium | | Mixed | |
|---|---|---|---|---|---|
| | | Mean | $\text{CVaR}_{0.1}$ | Mean | $\text{CVaR}_{0.1}$ |
| Cheetah | CQL | 33.2 | -15.0 | 214.1 | 12.0 |
| | ORAAC | **361.4** | **91.3** | 307.1 | 118.9 |
| | CODAC-N | 338 | -41 | 347.7 | 149.2 |
| | CODAC-C | 335 | -27 | **396.4** | **238.5** |
| Hopper | CQL | 877.9 | 693.0 | 189.2 | -21.4 |
| | ORAAC | 1007.1 | 767.6 | 876.3 | 524.9 |
| | CODAC-N | 993.7 | 952.5 | 1483.9 | **1457.6** |
| | CODAC-C | **1014.0** | **976.4** | **1551.2** | 1449.6 |
| Walker2d | CQL | 1524.3 | **1343.8** | 74.3 | -64.0 |
| | ORAAC | 1134.1 | 663.0 | 222.0 | -69.6 |
| | CODAC-N | **1537.3** | 1158.8 | 358.7 | 106.4 |
| | CODAC-C | 1120.8 | 902.3 | **450.0** | **261.4** |

| Dataset | BCQ | MOPO | CQL | ORAAC | CODAC |
|---|---|---|---|---|---|
| halfcheetah-random | 2.2 | **35.4** | 35.4 | 13.5 | 34.6 |
| hopper-random | 10.6 | **11.7** | 10.8 | 9.8 | 11.0 |
| walker2d-random | 4.9 | 13.6 | 7.0 | 3.2 | **18.7** |
| halfcheetah-medium | 40.7 | 42.3 | 44.4 | 41.0 | **46.3** |
| walker2d-medium | 53.1 | 17.8 | 79.2 | 27.3 | **82.0** |
| hopper-medium | 54.5 | 28.0 | 58.0 | 1.48 | **70.8** |
| halfcheetah-mixed | 38.2 | **53.1** | 46.2 | 30.0 | 44.1 |
| hopper-mixed | 33.1 | 67.5 | 48.6 | 16.3 | **100.2** |
| walker2d-mixed | 15.0 | **39.0** | 26.7 | 28 | 33.2 |
| halfcheetah-med-exp | 64.7 | 63.3 | 62.4 | 24.0 | **70.4** |
| walker2d-med-exp | 57.5 | 44.6 | 98.7 | 28.2 | **106.0** |
| hopper-med-exp | 110.9 | 23.7 | 111.0 | 18.2 | **112.0** |

region before proceeding to the goal; in contrast, all other agents traverse the risky region. For Ant, we include plots of the trajectories of trained agents in Appendix C.1 and videos in the supplement.

## 5.2 Risk-sensitive D4RL

**Tasks.** Next, we consider stochastic D4RL [43]. The original D4RL benchmark [10] consists of datasets collected by SAC agents of varying performance (Mixed, Medium, and Expert) on the Hopper, Walker2d, and HalfCheetah MuJoCo environments [41]; stochastic D4RL relabels the rewards to represent stochastic robot damage for behaviors such as unnatural gaits or high velocities; see Appendix C.2. The Expert dataset consists of rollouts from a fixed SAC agent trained to convergence; the Medium dataset is constructed the same way except the agent is trained to only achieve 50% of the expert agent's return. The Mixed dataset is the replay buffer of the Medium agent.

**Results.** In Table 2 (Left), we report the mean and $\text{CVaR}_{0.1}$ returns on test episodes from each approach, averaged over 5 random seeds. We show results on the Expert dataset in Appendix C.2. CODAC still achieves the strongest performance. As can be seen, CODAC-C and CODAC-N outperform both CQL and ORAAC on most datasets. Surprisingly, CODAC-N is quite effective on the $\text{CVaR}_{0.1}$ metric despite its risk-neutral objective; a likely explanation is that for these datasets, mean and CVaR performance are highly correlated. Furthermore, we observe that directly optimizing CVaR may lead to unstable training, potentially since CVaR estimates have higher variance. This instability occurs for both CODAC-C and ORAAC—on Walker2d-Medium, they perform worse than the risk-neutral algorithms. Overall, CODAC-C outperforms CODAC-N in terms of $\text{CVaR}_{0.1}$ on about half of the datasets, and often improves mean performance as well. Next, while ORAAC is generally effective on Medium datasets, it performs poorly on Mixed datasets; these results mirror the ones in Section 5.1. Finally, CQL's performance varies drastically across datasets; we hypothesize that learning the full distribution helps stabilize training in CODAC. In Appendix C.2, we also qualitatively analyze the behavior learned by CODAC compared to the baselines, demonstrating that the better CVaR performance CODAC obtains indeed translates to safer locomotion behaviors.

## 5.3 Risk-neutral D4RL

**Task.** Next, we show that CODAC is effective even when the goal is to optimize the standard expected return. To this end, we evaluate CODAC-N on the popular D4RL Mujoco benchmark [10].

**Baselines.** We compare to state-of-art algorithms benchmarked in [10] and [48], including Batch-Constrained Q-Learning (BCQ), Model-Based Offline Policy Optimization (MOPO) [48], and CQL. We also include ORAAC as an offline distributional RL baseline. We have omitted less competitive baselines included in [10] from the main text; a full comparison is included in Appendix C.3.

**Results.** Results for non-distributional approaches are directly taken from [10]; for ORAAC and CODAC, we evaluate them using 10 test episodes in the environment, averaged over 5 random seeds. As shown in Table 2 (Right), CODAC achieves strong performance across all 12 datasets, obtaining state-of-art results on 5 datasets (walker2d-random, hopper-medium, hopper-mixed, halfcheetah-medium-expert, and walker2d-medium-expert), demonstrating that performance improvements from

distributional learning also apply in the offline setting. Note that CODAC's advantage is not solely due to distributional RL—ORAAC also uses distributional RL, but in most cases underperforms prior state-of-the-art, These results suggest that CODAC's use of a conservative penalty is critical for it to achieve strong performance.

## 5.4 Analysis of Theoretical Insights

We perform additional experiments to validate that our theoretical insights in Section 3 hold in practice, suggesting that they help explain CODAC's empirical performance.

**Lower bound.** We show that in practice, CODAC obtains conservative estimates of the $Q$ and CVaR objectives across different dataset types (i.e., Medium vs. Mixed vs. Medium-Expert). Given an initial state $s_0$, we obtain a Monte Carlo (MC) estimate of $Q$ and CVaR for $(s_0, \pi(s_0))$ based on sampled rollouts from $s_0$, and compare them to the values predicted by the critic. In Table 3, we show results averaged over 10 random $s_0$ and with 100 MC samples for each $s_0$. CODAC obtains conservative estimates for both $Q$ and CVaR; in contrast, ORAAC overestimates these values, especially on Mixed datasets, and CQL only obtains conservative estimates for $Q$, not CVaR.

Table 3: **Monte-Carlo estimate vs. critic prediction.** The CODAC-predicted expected and $\text{CVaR}_{0.1}$ return is a lower bound on a MC estimate of the true value.

| Regular | Walker2d-Medium | | Walker2d-Mixed | | Walker2d-Medium-Expert | |
|---|---|---|---|---|---|---|
| | MC Return | Q-Estimate | MC Return | Q-Estimate | MC Return | Q-Estimate |
| CODAC | 240.2 | 55.7 | 127.1 | 97.6 | 370. | 39.7 |
| CQL | 247.2 | 53.0 | 124.5 | -45.2 | 369.7 | 116.4 |
| ORAAC | 245.2 | 302.2 | 118.2 | $7.70 \times 10^5$ | 68.2 | 322.2 |

| Stochastic | Walker2d-Medium | | Walker2d-Mixed | | Walker2d-Medium-Expert | |
|---|---|---|---|---|---|---|
| | MC $\text{CVaR}_{0.1}$ | Z-Estimate | MC $\text{CVaR}_{0.1}$ | Z-Estimate | MC $\text{CVaR}_{0.1}$ | Z-Estimate |
| CODAC | 185.7 | 204.2 | 85.6 | 59.9 | 265.3 | -127.8 |
| ORAAC | 201.9 | 367.6 | 50.9 | $1.54 \times 10^6$ | 199.5 | 343.5 |

**Gap-Expansion.** Next, we verify that CODAC's quantile estimates expand their gap between in-distribution and out-of-distribution actions. We use the D4RL Medium-Expert datasets where CODAC uniformly performs well, making them ideal for understanding the source of CODAC's empirical performance. We train "CODAC w.o. Penalty", a non-conservative variant of CODAC (i.e., $\alpha = 0$), and use its actor as $\mu$ and its critic as $F_{Z^\pi}^{-1}$. Next, for each dataset, we randomly sample 1000 state-action pairs and 32 quantiles $\tau$, resulting in 32000 $(s, a, \tau)$ tuples; for each one, we compute the quantile gaps for CODAC and CODAC w.o. Penalty. In Table 4, we show the percentage of tuples where each CODAC variant has a larger quantile gap, along with their average return. As can be seen, CODAC has a larger gap for more than $90\%$ of the tuples on all datasets, as well as significantly higher returns. These results show that gap-expansion holds in practice and suggest that it helps CODAC achieve good performance.

Table 4: **Gap-expansion:** CODAC expands the quantile gap and obtains higher returns than an ablation without the conservative penalty.

| | HalfCheetah-Medium-Expert | | Hopper-Medium-Expert | | Walker2d-Medium-Expert | |
|---|---|---|---|---|---|---|
| | Positive Gap % | Return | Positive Gap % | Return | Positive Gap % | Return |
| CODAC | **95.3** | **93.6** | **91.3** | **111.9** | **91.1** | **111.3** |
| CODAC w.o. Penalty | 4.7 | 12.1 | 8.7 | 25.8 | 8.9 | 5.9 |

## 6 Conclusion

We have introduced Conservative Offline Distributional Actor-Critic (CODAC), a general purpose offline distributional reinforcement learning algorithm. We have proven that CODAC obtains conservative estimates of the return quantile, which translate into lower bounds on $Q$ and CVaR values. In our experiments, CODAC outperforms prior approaches on both stochastic, risk-sensitive offline RL benchmarks, as well as traditional, risk-neutral benchmarks.

One limitation of our work is that CODAC has hyperparameters that must be tuned (in particular, the penalty magnitude $\alpha$). As in prior work, we choose these hyperparameters by evaluate online rollouts in the environment. Designing better hyperparameter selection strategies for offline RL is an important direction for future work. Finally, we do not foresee any societal impacts or ethical concerns for our work, other than the usual risks around algorithms for improving robotics capabilities.

## Acknowledgments and Disclosure of Funding

This work is funded in part by an Amazon Research Award, gift funding from NEC Laboratories America, NSF Award CCF-1910769, and ARO Award W911NF-20-1-0080. The U.S. Government is authorized to reproduce and distribute reprints for Government purposes notwithstanding any copyright notation herein.

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
