# A Proofs

## A.1 Proof of Lemma 3.4

Recall that the $p$-Wasserstein distance is the $L_p$ metric between quantile functions (see Eq. 3). Thus, we can re-write the CODAC objective as

$$\alpha \cdot \mathbb{E}_{U(\tau), \mathcal{D}(s,a)} \left[ c_0(s,a) \cdot F_{Z(s,a)}^{-1}(\tau) \right] + \mathbb{E}_{\mathcal{D}(s,a)} \int_0^1 \left| F_{Z(s,a)}^{-1}(\tau) - F_{\hat{\mathcal{T}}^\pi \hat{Z}^k(s,a)}^{-1}(\tau) \right|^p d\tau$$

$$= \int_0^1 \mathbb{E}_{\mathcal{D}(s,a)} \left[ \alpha \cdot c_0(s,a) \cdot F_{Z(s,a)}^{-1}(\tau) + \left| F_{Z(s,a)}^{-1}(\tau) - F_{\hat{\mathcal{T}}^\pi \hat{Z}^k(s,a)}^{-1}(\tau) \right|^p \right] d\tau.$$

We consider a perturbation

$$G_{s,a}^\epsilon(\tau) = F_{Z(s,a)}^{-1}(\tau) + \epsilon \cdot \phi_{s,a}(\tau)$$

for arbitrary smooth functions $\phi_{s,a}$ with compact support $[V_{\min}, V_{\max}]$, yielding

$$\int_0^1 \mathbb{E}_{\mathcal{D}(s,a)} \left[ \alpha c_0(s,a) \cdot G_{s,a}^\epsilon(\tau) + \left| G_{s,a}^\epsilon(\tau) - F_{\hat{\mathcal{T}}^\pi \hat{Z}^k(s,a)}^{-1}(\tau) \right|^p \right] d\tau.$$

Taking the derivative with respect to $\epsilon$ at $\epsilon = 0$, we have

$$\frac{d}{d\epsilon} \int_0^1 \mathbb{E}_{\mathcal{D}(s,a)} \left[ \alpha c_0(s,a) \cdot G_{s,a}^\epsilon(\tau) + \left| G_{s,a}^\epsilon(\tau) - F_{\hat{\mathcal{T}}^\pi \hat{Z}^k(s,a)}^{-1}(\tau) \right|^p \right] d\tau \bigg|_{\epsilon=0}$$

$$= \mathbb{E}_{\mathcal{D}(s,a)} \int_0^1 \left[ \alpha c_0(s,a) + p \left| F_{Z(s,a)}^{-1}(\tau) - F_{\hat{\mathcal{T}}^\pi \hat{Z}^k(s,a)}^{-1}(\tau) \right|^{p-1} \text{sign}\left( F_{Z(s,a)}^{-1}(\tau) - F_{\hat{\mathcal{T}}^\pi \hat{Z}^k(s,a)}^{-1}(\tau) \right) \right] \phi_{s,a}(\tau) d\tau.$$

This term must equal $0$ for $F_{Z(s,a)}^{-1}$ to minimize the objective; otherwise, some perturbation $G_{s,a}^\epsilon$ decreases the objective value. Since $\phi_{s,a}$ are arbitrary, it must equal zero for each $s, a$ individually; otherwise, increasing $\phi_{s,a}$ would increase the term, making it nonzero. Thus, we have

$$\int_0^1 \left[ \alpha c_0(s,a) + p \left| F_{Z(s,a)}^{-1}(\tau) - F_{\hat{\mathcal{T}}^\pi \hat{Z}^k(s,a)}^{-1}(\tau) \right|^{p-1} \text{sign}\left( F_{Z(s,a)}^{-1}(\tau) - F_{\hat{\mathcal{T}}^\pi \hat{Z}^k(s,a)}^{-1}(\tau) \right) \right] \phi_{s,a}(\tau) d\tau = 0$$

for all $s, a$. Then, by the fundamental lemma of the calculus of variations, for each $s, a$, if this term is zero for all $\phi_{s,a}$, then the integrand must be zero—i.e.,

$$\alpha c_0(s,a) + p \left| F_{Z(s,a)}^{-1}(\tau) - F_{\hat{\mathcal{T}}^\pi \hat{Z}^k(s,a)}^{-1}(\tau) \right|^{p-1} \text{sign}\left( F_{Z(s,a)}^{-1}(\tau) - F_{\hat{\mathcal{T}}^\pi \hat{Z}^k(s,a)}^{-1}(\tau) \right) = 0,$$

which holds if and only if

$$F_{Z(s,a)}^{-1}(\tau) = F_{\hat{\mathcal{T}}^\pi \hat{Z}^k(s,a)}^{-1}(\tau) - c(s,a).$$

where $c(s,a) = |\alpha p^{-1} c_0(s,a)|^{1/(p-1)} \cdot \text{sign}(c_0(s,a))$, Clearly, this choice of $Z$ is valid, so the claim follows. $\square$

## A.2 Proof of Theorem 3.6

First, we have the following result, which is a concentration bound on the quantile values; this result enables us to bound the estimation error of $\hat{\mathcal{T}}^\pi$ compared to $\mathcal{T}^\pi$:

**Lemma A.1.** *Let* $n(s,a) = |\{(s,a) \mid (s,a,r,s') \in \mathcal{D}\}|$ *be the number of times* $(s,a)$ *occurs in* $\mathcal{D}$. *For any return distribution* $Z$ *with* $\zeta$-*strongly monotone CDF* $F_{Z(s,a)}$ *and any* $\delta \in \mathbb{R}_{>0}$, *with probability at least* $1 - \delta$, *for all* $s \in \mathcal{D}$ *and* $a \in \mathcal{A}$, *we have*

$$\| F_{\hat{\mathcal{T}}^\pi Z(s,a)}^{-1} - F_{\mathcal{T}^\pi Z(s,a)}^{-1} \|_\infty \leq \Delta(s,a) \quad \text{where} \quad \Delta(s,a) = \frac{1}{\zeta} \sqrt{\frac{5|\mathcal{S}|}{n(s,a)} \log \frac{4|\mathcal{S}||\mathcal{A}|}{\delta}}.$$

This lemma follows by first using the Dvoretzky-Kiefer-Wolfowitz inequality to bound the error of the empirical CDF $F_{\tilde{\mathcal{T}}^\pi Z(s,a)}$ compared to the true CDF $F_{\mathcal{T}^\pi Z(s,a)}$ using similar analysis as in [17], and then leveraging monotonicity to bound the quantile functions; we give a proof in Appendix A.4. Next, we have the following key lemma, which relates one-step distributional Bellman contraction to an $\infty$-norm bound at the fixed point.

**Lemma A.2.** *If $Z$ satisfies $\|F_{Z(s,a)}^{-1} - F_{\mathcal{T}Z(s,a)}^{-1}\|_\infty \le \beta$ for all $s \in \mathcal{S}$ and $a \in \mathcal{A}$, then*

$$\|F_{Z(s,a)}^{-1} - F_{Z^\pi(s,a)}^{-1}\|_\infty \le (1-\gamma)^{-1}\beta \qquad (\forall s \in \mathcal{S}, a \in \mathcal{A}),$$

We give a proof in Appendix A.5. As we discuss in Appendix A.6, we can use this result to obtain bounds on the fixed point of the non-conservative empirical Bellman operator $\hat{\mathcal{T}}$. Now, we prove Theorem 3.6. First, with probability at least $1 - \delta$, we have

$$\begin{aligned}
F_{\tilde{\mathcal{T}}^\pi Z^\pi(s,a)}^{-1}(\tau) &= F_{\hat{\mathcal{T}}^\pi Z^\pi(s,a)}^{-1}(\tau) - c(s,a) \\
&\le F_{\mathcal{T}^\pi Z^\pi(s,a)}^{-1}(\tau) - c(s,a) + \Delta(s,a) \\
&= F_{Z^\pi(s,a)}^{-1}(\tau) - c(s,a) + \Delta(s,a),
\end{aligned} \tag{8}$$

where the first step follows by Lemma 3.4 (noting that it holds for arbitrary $\tilde{Z}^k$, and substituting $\tilde{Z}^k = Z^\pi$), the second step holds with probability at least $1 - \delta$ by Lemma A.1 with $Z = Z^\pi$ (since $Z^\pi$ is $\zeta$-strongly monotone), and the third step follows since $Z^\pi = \mathcal{T}^\pi Z^\pi$ is the fixed point of $\mathcal{T}^\pi$.

Now, rearranging (8), we have

$$\begin{aligned}
F_{Z^\pi(s,a)}^{-1}(\tau) &\ge F_{\tilde{\mathcal{T}}^\pi Z^\pi(s,a)}^{-1}(\tau) + c(s,a) - \Delta(s,a) \\
&\ge F_{\tilde{\mathcal{T}}^\pi Z^\pi(s,a)}^{-1}(\tau) + \min_{s,a}\{c(s,a) - \Delta(s,a)\} \\
&\ge F_{\tilde{Z}^\pi(s,a)}^{-1}(\tau) + (1-\gamma)^{-1}\min_{s,a}\{c(s,a) - \Delta(s,a)\},
\end{aligned} \tag{9}$$

where in the last step, we have applied Lemma A.6 for the case $\ge$ and $\tilde{\mathcal{T}}^\pi$, and with $\beta = \min_{s,a}\{c(s,a) - \Delta(s,a)\}$. Finally, note that for the last term in (9) to be positive, we need

$$\alpha p^{-1} c_0(s,a) \ge \Delta(s,a)^{p-1} \qquad (\forall s,a).$$

Since we have assumed that $c_0(s,a) > 0$, this expression is in turn equivalent to

$$\alpha \ge \max_{s,a}\left\{\frac{p \cdot \Delta(s,a)^{p-1}}{c_0(s,a)}\right\},$$

so the claim holds. $\square$

## A.3 Proof of Theorem 3.8

**Lemma A.3.** *For any $Z$ and any $\bar{\Delta}$, for sufficiently large $\alpha$, with probability at least $1 - \delta$, we have*

$$\mathbb{E}_{\hat{\pi}_\beta(a|s)} F_{\tilde{\mathcal{T}}^\pi Z(s,a)}^{-1}(\tau) - \mathbb{E}_{\mu(a|s)} F_{\tilde{\mathcal{T}}^\pi Z(s,a)}^{-1}(\tau) \ge \mathbb{E}_{\hat{\pi}_\beta(a|s)} F_{\mathcal{T}^\pi Z(s,a)}^{-1}(\tau) - \mathbb{E}_{\mu(a|s)} F_{\mathcal{T}^\pi Z(s,a)}^{-1}(\tau) + \bar{\Delta}.$$

*Proof.* First, by Lemma 3.4, we have

$$F_{\tilde{\mathcal{T}}^\pi Z(s,a)}^{-1}(\tau) = F_{\hat{\mathcal{T}}^\pi Z(s,a)}^{-1}(\tau) - c(s,a).$$

Then, by Lemma A.1, with probability at least $1 - \delta$, we have

$$F_{\mathcal{T}^\pi Z(s,a)}^{-1}(\tau) - c(s,a) - \Delta(s,a) \le F_{\tilde{\mathcal{T}}^\pi Z(s,a)}^{-1}(\tau) \le F_{\mathcal{T}^\pi Z(s,a)}^{-1}(\tau) - c(s,a) + \Delta(s,a).$$

Taking the expectation over $\hat{\pi}_\beta$ (resp., $\mu$) of the lower (resp., upper) bound gives

$$\begin{aligned}
\mathbb{E}_{\hat{\pi}_\beta(a|s)} F_{\tilde{\mathcal{T}}^\pi Z(s,a)}^{-1}(\tau) &\ge \mathbb{E}_{\hat{\pi}_\beta(a|s)} F_{\mathcal{T}^\pi Z(s,a)}^{-1}(\tau) - \mathbb{E}_{\hat{\pi}_\beta(a|s)} c(s,a) - \mathbb{E}_{\hat{\pi}_\beta(a|s)} \Delta(s,a) \\
\mathbb{E}_{\mu(a|s)} F_{\tilde{\mathcal{T}}^\pi Z(s,a)}^{-1}(\tau) &\le \mathbb{E}_{\mu(a|s)} F_{\mathcal{T}^\pi Z(s,a)}^{-1}(\tau) - \mathbb{E}_{\mu(a|s)} c(s,a) + \mathbb{E}_{\mu(a|s)} \Delta(s,a),
\end{aligned}$$

respectively. Recall that $p = 2$. Then, subtracting the latter from the former and rearranging terms,

$$\mathbb{E}_{\hat{\pi}_\beta(a|s)} F^{-1}_{\tilde{\mathcal{T}}^\pi Z(s,a)}(\tau) - \mathbb{E}_{\mu(a|s)} F^{-1}_{\tilde{\mathcal{T}}^\pi Z(s,a)}(\tau) \geq \mathbb{E}_{\hat{\pi}_\beta(a|s)} F^{-1}_{\mathcal{T}^\pi Z(s,a)}(\tau) - \mathbb{E}_{\mu(a|s)} F^{-1}_{\mathcal{T}^\pi Z(s,a)}(\tau)$$
$$+ (\alpha/2)\bar{c}(s) - \bar{\Delta}(s),$$

where

$$\bar{c}(s) = \mathbb{E}_{\mu(a|s)} c_0(s,a) - \mathbb{E}_{\hat{\pi}_\beta(a|s)} c_0(s,a)$$
$$\bar{\Delta}(s) = \mathbb{E}_{\mu(a|s)} \Delta(s,a) + \mathbb{E}_{\hat{\pi}_\beta(a|s)} \Delta(s,a).$$

Note that to show the claim, it suffices to show that for sufficient large $\alpha$, we have

$$(\alpha/2)\bar{c}(s) \geq \bar{\Delta}(s) + \bar{\Delta} \qquad (\forall s). \tag{10}$$

To this end, note that

$$\mathbb{E}_{\hat{\pi}_\beta(a|s)} c(s,a) = \sum_a (\mu(a \mid s) - \hat{\pi}_\beta(a \mid s)) = 0,$$

and

$$\mathbb{E}_{\mu(a|s)} c(s,a)$$
$$= \sum_a \left( \frac{\mu(a \mid s) - \hat{\pi}_\beta(a \mid s)}{\hat{\pi}_\beta(a \mid s)} \right) \mu(a \mid s)$$
$$= \sum_a \left( \frac{\mu(a \mid s) - \hat{\pi}_\beta(a \mid s)}{\hat{\pi}_\beta(a \mid s)} \right) (\mu(a \mid s) - \hat{\pi}_\beta(a \mid s)) + \sum_a \left( \frac{\mu(a \mid s) - \hat{\pi}_\beta(a \mid s)}{\hat{\pi}_\beta(a \mid s)} \right) \hat{\pi}_\beta(a \mid s)$$
$$= \sum_a \left( \frac{\mu(a \mid s) - \hat{\pi}_\beta(a \mid s)}{\hat{\pi}_\beta(a \mid s)} \right) (\mu(a \mid s) - \hat{\pi}_\beta(a \mid s))$$
$$= \sum_a \frac{(\mu(a \mid s) - \hat{\pi}_\beta(a \mid s))^2}{\hat{\pi}_\beta(a \mid s)},$$

so we have

$$\bar{c}(s) = \sum_a \frac{(\mu(a \mid s) - \hat{\pi}_\beta(a \mid s))^2}{\hat{\pi}_\beta(a \mid s)} = \text{Var}_{\hat{\pi}_\beta(a|s)} \left[ \frac{\mu(a \mid s) - \hat{\pi}_\beta(a \mid s)}{\hat{\pi}_\beta(a \mid s)} \right] > 0,$$

where the last inequality holds since $\mu(a \mid s) \neq \hat{\pi}_\beta(a \mid s)$. Thus, for (10) to hold, it suffices to have

$$\alpha \geq 2 \cdot \max_s \left\{ \text{Var}_{\hat{\pi}_\beta(a|s)} \left[ \frac{\mu(a \mid s) - \hat{\pi}_\beta(a \mid s)}{\hat{\pi}_\beta(a \mid s)} \right]^{-1} \cdot (\bar{\Delta}(s) + \bar{\Delta}) \right\}.$$

The claim follows. $\qquad\square$

Now, let $Z_0 = \tilde{Z}_0$, and let $Z_k = (\mathcal{T}^\pi)^k Z_0$ and $\tilde{Z}_k = (\tilde{\mathcal{T}}^\pi)^k \tilde{Z}_0$. Applying Lemma A.3 with $Z = \tilde{Z}^k$ and $\bar{\Delta} = 4V_{\max}$, we have

$$\mathbb{E}_{\hat{\pi}_\beta(a|s)} F^{-1}_{\tilde{\mathcal{T}}^\pi \tilde{Z}^k(s,a)}(\tau) - \mathbb{E}_{\mu(a|s)} F^{-1}_{\tilde{\mathcal{T}}^\pi \tilde{Z}^k(s,a)}(\tau)$$
$$\geq \mathbb{E}_{\hat{\pi}_\beta(a|s)} F^{-1}_{\mathcal{T}^\pi \tilde{Z}^k(s,a)}(\tau) - \mathbb{E}_{\mu(a|s)} F^{-1}_{\mathcal{T}^\pi \tilde{Z}^k(s,a)}(\tau) + \bar{\Delta}$$
$$= \mathbb{E}_{\hat{\pi}_\beta(a|s)} F^{-1}_{\mathcal{T}^\pi Z^k(s,a)}(\tau) - \mathbb{E}_{\mu(a|s)} F^{-1}_{\mathcal{T}^\pi Z^k(s,a)}(\tau) + \bar{\Delta}$$
$$+ \left( \mathbb{E}_{\hat{\pi}_\beta(a|s)} F^{-1}_{\mathcal{T}^\pi \tilde{Z}^k(s,a)}(\tau) - \mathbb{E}_{\mu(a|s)} F^{-1}_{\mathcal{T}^\pi \tilde{Z}^k(s,a)}(\tau) \right)$$
$$- \left( \mathbb{E}_{\hat{\pi}_\beta(a|s)} F^{-1}_{\mathcal{T}^\pi Z^k(s,a)}(\tau) - \mathbb{E}_{\mu(a|s)} F^{-1}_{\mathcal{T}^\pi Z^k(s,a)}(\tau) \right)$$
$$\geq \mathbb{E}_{\hat{\pi}_\beta(a|s)} F^{-1}_{\mathcal{T}^\pi Z^k(s,a)}(\tau) - \mathbb{E}_{\mu(a|s)} F^{-1}_{\mathcal{T}^\pi Z^k(s,a)}(\tau)$$
$$+ \bar{\Delta} - 4V_{\max}$$
$$= \mathbb{E}_{\hat{\pi}_\beta(a|s)} F^{-1}_{\mathcal{T}^\pi Z^k(s,a)}(\tau) - \mathbb{E}_{\mu(a|s)} F^{-1}_{\mathcal{T}^\pi Z^k(s,a)}(\tau).$$

The claim follows by taking the limit $k \to \infty$. $\quad\square$

## A.4 Proof of Lemma A.1

We first prove a bound on the concentration of the empirical CDF to the true CDF. A similar result has been previously derived in [17]; our proof is based on theirs.

**Lemma A.4.** *For all $\delta \in \mathbb{R}_{>0}$, with probability at least $1 - \delta$, for any $Z \in \mathcal{Z}$, for all $(s, a) \in \mathcal{D}$,*

$$\|F_{\hat{\mathcal{T}}^\pi Z(s,a)} - F_{\mathcal{T}^\pi Z(s,a)}\|_\infty \leq \sqrt{\frac{5|\mathcal{S}|}{n(s,a)} \log \frac{4|\mathcal{S}||\mathcal{A}|}{\delta}} \tag{11}$$

*Proof.* By the definition of distributional Bellman operator applied to the CDF function, we have that

$$F_{\hat{\mathcal{T}}^\pi Z(s,a)}(x) - F_{\mathcal{T}^\pi Z(s,a)}(x)$$
$$= \sum_{s',a'} \hat{P}(s' \mid s,a)\pi(a' \mid s')F_{\gamma Z(s',a')+\hat{R}(s,a)}(x) - \sum_{s',a'} P(s' \mid s,a)\pi(a' \mid s')F_{\gamma Z(s',a')+R(s,a)}(x).$$

Adding and subtracting $\sum_{s',a'} \hat{P}(s' \mid s,a)\pi(a' \mid s')F_{\gamma Z(s',a')+R(s,a)}(x)$ from this expression gives

$$\sum_{s',a'} \hat{P}(s' \mid s,a)\pi(a' \mid s')\Big(F_{\gamma Z(s',a')+\hat{R}(s,a)}(x) - F_{\gamma Z(s',a')+R(s,a)}(x)\Big)$$

$$+ \sum_{s',a'} \Big(\hat{P}(s' \mid s,a) - P(s' \mid s,a)\Big)\pi(a' \mid s')F_{\gamma Z(s',a')+R(s,a)}(x).$$

We proceed by bounding the two terms in the summation. For the first term, observe that

$$F_{\gamma Z(s',a')+\hat{R}(s,a)}(x) - F_{\gamma Z(s',a')+R(s,a)}(x)$$
$$= \int \Big[F_{\hat{R}(s,a)}(r) - F_{R(s,a)}(r)\Big]dF_{\gamma Z(s',a')}(x - r)$$
$$\leq \int \Big|F_{\hat{R}(s,a)}(r) - F_{R(s,a)}(r)\Big|dF_{\gamma Z(s',a')}(x - r)$$
$$\leq \sup_r \Big|F_{\hat{R}(s,a)}(r) - F_{R(s,a)}(r)\Big| \int dF_{\gamma Z(s',a')}(x - r)$$
$$= \Big\|F_{\hat{R}(s,a)}(r) - F_{R(s,a)}(r)\Big\|_\infty.$$

Therefore, we have

$$\sum_{s',a'} \hat{P}(s' \mid s,a)\pi(a' \mid s')\Big(F_{\gamma Z(s',a')+\hat{R}(s,a)}(x) - F_{\gamma Z(s',a')+R(s,a)}(x)\Big)$$

$$\leq \sum_{s',a'} \hat{P}(s' \mid s,a)\pi(a' \mid s')\Big\|F_{\hat{R}(s,a)}(r) - F_{R(s,a)}(r)\Big\|_\infty$$

$$= \Big\|F_{\hat{R}(s,a)}(r) - F_{R(s,a)}(r)\Big\|_\infty$$

The second term can be bounded as follows:

$$\sum_{s',a'} \Big(\hat{P}(s' \mid s,a) - P(s' \mid s,a)\Big)\pi(a' \mid s')F_{\gamma Z(s',a')+R(s,a)}(x)$$

$$= \sum_{s'} \Big(\hat{P}(s' \mid s,a) - P(s' \mid s,a)\Big)\sum_{a'}\pi(a' \mid s')F_{\gamma Z(s',a')+R(s,a)}(x)$$

$$\leq \Big\|\hat{P}(\cdot \mid s,a) - P(\cdot \mid s,a)\Big\|_1 \cdot \Big\|\sum_{a'}\pi(a' \mid \cdot)F_{\gamma Z(\cdot,a')+R(s,a)}(x)\Big\|_\infty$$

$$\leq \Big\|\hat{P}(\cdot \mid s,a) - P(\cdot \mid s,a)\Big\|_1 \cdot \Big\|\sum_{a'}\pi(a' \mid \cdot)\Big\|_\infty$$

$$= \Big\|\hat{P}(\cdot \mid s,a) - P(\cdot \mid s,a)\Big\|_1.$$

Together, we have

$$\left| F_{\hat{\mathcal{T}}^\pi Z(s,a)}(x) - F_{\mathcal{T}^\pi Z(s,a)}(x) \right| \le \left\| F_{\hat{R}(s,a)}(r) - F_{R(s,a)}(r) \right\|_\infty + \left\| \hat{P}(s' \mid s, a) - P(s' \mid s, a) \right\|_1 .$$

Finally, the inequalities can be bounded using the Dvoretzky–Kiefer–Wolfowitz (DKW) inequality and the Hoeffding's inequality, giving us the desired results. By the DKW inequality, we have that with probability $1 - \delta/2$, for all $(s, a) \in \mathcal{D}$,

$$\left\| F_{\hat{R}(s,a)}(r) - F_{R(s,a)}(r) \right\|_\infty \le \sqrt{\frac{1}{2n(s,a)} \ln \frac{4|\mathcal{S}||\mathcal{A}|}{\delta}}$$

Similarly, by Hoeffding's inequality and an $\ell_1$ concentration bound for multinomial distribution[3], we have

$$\max_{s,a} \left\| \hat{P}(\cdot \mid s, a) - P(\cdot \mid s, a) \right\|_1 \le \sqrt{\frac{2|\mathcal{S}|}{n(s,a)} \ln \frac{4|\mathcal{S}||\mathcal{A}|}{\delta}}$$

The claim follows by combining the two inequalities. $\qquad\square$

Next, we prove a general result that translates bounds on CDFs into bounds on quantile functions.

**Lemma A.5.** *Consider two CDFs $F$ and $G$ with support $\mathcal{X}$. Suppose that $F$ is $\zeta$-strongly monotone and that $\|F - G\|_\infty \le \epsilon$. Then, $\|F^{-1} - G^{-1}\|_\infty \le \epsilon/\zeta$.*

*Proof.* First, note that

$$F^{-1}(y) - G^{-1}(y) = \int_{G^{-1}(y)}^{F^{-1}(y)} dx = \int_{F(G^{-1}(y))}^{y} dF^{-1}(y'),$$

where the first equality follows by fundamental theorem of calculus, and the second by a change of variable $y' = F(x)$. Since $F(F^{-1}(y')) = y'$, we have $F'(F^{-1}(y')) dF^{-1}(y') = dy'$, so

$$dF^{-1}(y') = \frac{dy'}{F'(F^{-1}(y'))} \le \frac{dy'}{\zeta},$$

where the inequality follows by $\zeta$-strong monotonicity. As a consequence, we have

$$\int_{F(G^{-1}(y))}^{y} dF^{-1}(y') \le \int_{F(G^{-1}(y))}^{y} \frac{dy'}{\zeta} = \frac{(y - F(G^{-1}(y)))}{\zeta} = \frac{G(G^{-1}(y)) - F(G^{-1}(y))}{\zeta} \le \frac{\epsilon}{\zeta},$$

where the last inequality follows since $\|G - F\|_\infty \le \epsilon$. The claim follows. $\qquad\square$

Finally, Lemma A.1 follows by substituting $F = F_{\hat{\mathcal{T}}^\pi Z(s,a)}(x)$, $G = F_{\mathcal{T}^\pi Z(s,a)}(x)$, and $\epsilon = \sqrt{\frac{5|\mathcal{S}|}{n(s,a)} \log \frac{4|\mathcal{S}||\mathcal{A}|}{\delta}}$ into Lemma A.5, where the condition $\|F - G\|_\infty \le \epsilon$ holds by Lemma A.4. $\qquad\square$

## A.5  Proof of Lemma A.2

We prove the following slightly stronger result:

**Lemma A.6.** *For any $\beta \in \mathbb{R}$, if $Z$ satisfies*

$$F_{Z(s,a)}^{-1}(\tau) \ge F_{\mathcal{T}^\pi Z(s,a)}^{-1}(\tau) + \beta \qquad (\forall \tau \in [0, 1]) \tag{12}$$

*for all $s \in \mathcal{S}$ and $a \in \mathcal{A}$, then we have*

$$F_{Z(s,a)}^{-1}(\tau) \ge F_{Z^\pi(s,a)}^{-1}(\tau) + (1 - \gamma)^{-1}\beta \qquad (\forall \tau \in [0, 1]).$$

*The result holds with $\ge$ replaced by $\le$, or with $\mathcal{T}^\pi$ and $Z^\pi$ replaced by $\hat{\mathcal{T}}^\pi$ and $\hat{Z}^\pi$ or $\tilde{\mathcal{T}}^\pi$ and $\tilde{Z}^\pi$.*

---

[3]See https://nanjiang.cs.illinois.edu/files/cs598/note3.pdf for a derivation.

*Proof.* We prove the first case; the cases with $\geq$, and the cases with $\hat{\mathcal{T}}^\pi$ and $\hat{Z}^\pi$ follow by the same argument. First, we show that

$$F_{\mathcal{T}^\pi Z(s,a)}(x) \geq F_{Z(s,a)}(x + \beta) \qquad (\forall x \in [V_{\min}, V_{\max}]). \qquad (13)$$

To this end, note that rearranging (12), we have

$$F_{\mathcal{T}^\pi Z(s,a)}(F_{Z(s,a)}^{-1}(\tau) - \beta) \geq \tau.$$

Then, substituting $\tau = F_{\hat{Z}^\pi(s,a)}(x + \beta)$ yields (13); note that such $\tau$ must exist since the CDF is defined on all of $\mathbb{R}$. Next, we show that

$$F_{\mathcal{T}^\pi Z(s,a)}^{-1}(\tau) \geq F_{\mathcal{T}^\pi(\mathcal{T}^\pi Z(s,a))}^{-1}(\tau) + \gamma\beta \qquad (\forall \tau \in [0,1]), \qquad (14)$$

where the parts changed from (12) are highlighted in red. Intuitively, this claim says that $\mathcal{T}^\pi$ distributes additively to the constant $\beta$, and since $\mathcal{T}^\pi$ is a $\gamma$-contraction in $\bar{d}_p$, we have $\mathcal{T}^\pi\beta \leq \gamma\beta$. To show (14), first note that

$$\begin{aligned}
F_{\mathcal{T}^\pi(\mathcal{T}^\pi Z(s,a))}(x) &= \sum_{s',a'} P^\pi(s', a' \mid s, a) \int F_{\mathcal{T}^\pi Z(s',a')} \left( \frac{x - r}{\gamma} \right) dF_{R(s,a)}(r) \\
&\geq \sum_{s',a'} P^\pi(s', a' \mid s, a) \int F_{Z(s',a')} \left( \frac{x - r}{\gamma} + \beta \right) dF_{R(s,a)}(r) \\
&= \sum_{s',a'} P^\pi(s', a' \mid s, a) \int F_{\gamma Z(s',a')}(x - r + \gamma\beta) dF_{R(s,a)}(r) \\
&= \sum_{s',a'} P^\pi(s', a' \mid s, a) F_{R(s,a)+\gamma Z(s',a')}(x + \gamma\beta) \\
&= F_{\mathcal{T}^\pi Z(s,a)}(x + \gamma\beta),
\end{aligned}$$

where the first step follows by derivation of the Bellman operator for the CDF, the second step follows from (13), and the third step follows from the property of a CDF function. It follows that

$$F_{\mathcal{T}^\pi Z(s,a)}^{-1}(F_{\mathcal{T}^\pi(\mathcal{T}^\pi Z(s,a))}(x)) \geq x + \gamma\beta.$$

Setting $\tau = F_{\mathcal{T}^\pi(\mathcal{T}^\pi Z(s,a))}(x)$, we have

$$F_{\mathcal{T}^\pi Z(s,a)}^{-1}(\tau) \geq F_{\mathcal{T}^\pi(\mathcal{T}^\pi Z(s,a))}^{-1}(\tau) + \gamma\beta$$

for all $\tau \in [0,1]$; thus, we have shown (14). Now, by induction on $\mathcal{T}^\pi$, we have

$$F_{(\mathcal{T}^\pi)^k Z(s,a)}^{-1}(\tau) \geq F_{(\mathcal{T}^\pi)^{k+1} Z(s,a)}^{-1}(\tau) + \gamma^k\beta$$

for all $k \in \mathbb{N}$. Summing these inequalities over $k \in \{0, 1, ..., n\}$ inequality gives

$$\sum_{k=0}^{n} F_{(\mathcal{T}^\pi)^k Z(s,a)}^{-1}(\tau) \geq \sum_{k=0}^{n} F_{(\mathcal{T}^\pi)^k(\mathcal{T}^\pi Z(s,a))}^{-1}(\tau) + \sum_{k=0}^{n} \gamma^k\beta$$

Subtracting common terms from both sides and evaluating the sum over $\gamma^k$, we have

$$F_{Z(s,a)}^{-1}(\tau) \geq F_{(\mathcal{T}^\pi)^{n+1} Z(s,a)}^{-1}(\tau) + \frac{1 - \gamma^{n+1}}{1 - \gamma}\beta.$$

Taking $n \to \infty$, we have

$$F_{Z(s,a)}^{-1}(\tau) \geq F_{Z^\pi(s,a)}^{-1}(\tau) - (1 - \gamma)^{-1}\beta,$$

where we have used the fact that $Z^\pi$ is the fixed point of $\mathcal{T}^\pi$. The claim follows. $\qquad \square$

### A.6 Bound on error of the fixed-point of the empirical distributional bellman operator

We can use our techniques to prove finite-sample bounds on the error of using value iteration with the empirical Bellman operator $\hat{\mathcal{T}}$ compared to the true Bellman operator $\mathcal{T}$.

**Theorem A.7.** *We have* $\|F_{\hat{Z}^\pi(s,a)}^{-1} - F_{Z^\pi(s,a)}\|_\infty \leq (1 - \gamma)^{-1}\Delta_{max}$, *where* $\hat{Z}^\pi$ *and* $Z^\pi$ *are the fixed-points of* $\hat{\mathcal{T}}^\pi$ *and* $\mathcal{T}^\pi$, *respectively.*

*Proof.* Let $\Delta_{\max} = \max_{s,a} \Delta(s, a)$. We have $\|F_{\hat{Z}^\pi(s,a)}^{-1} - F_{\mathcal{T}^\pi \hat{Z}^\pi(s,a)}\|_\infty \leq \Delta_{\max}$ by Lemma A.1 with $Z = \hat{Z}^\pi$. Thus, we have $\|F_{\hat{Z}^\pi(s,a)}^{-1} - F_{Z^\pi(s,a)}\|_\infty \leq (1 - \gamma)^{-1}\Delta_{\max}$ by Lemma A.2. $\qquad \square$

# B Algorithm and implementation details

In this section, we describe our practical implementation of CODAC in detail.

## B.1 Actor-Critic objective

We first describe a modification to the CODAC objective, which admits *learnable* $\alpha$, instead of having to fix it to a constant value throughout the entirety of training. Recall that the original objective is

$$\hat{Z}^{k+1} = \arg\min_{Z} \left\{ \alpha \cdot \mathbb{E}_{U(\tau)} \left[ \mathbb{E}_{\mathcal{D}(s)} \log \sum_{a} \exp(F_{Z(s,a)}^{-1}(\tau)) - \mathbb{E}_{\mathcal{D}(s,a)} F_{Z(s,a)}^{-1}(\tau) \right] + \mathcal{L}_p(Z, \hat{\mathcal{T}}^{\pi^k} \hat{Z}^k) \right\},$$

We first provide a derivation of the above objective; this portion largely follows from [21]. We first introduce a *regularization* term $\mathcal{R}(\mu)$ to obtain a well-defined optimization problem:

$$\hat{Z}^{k+1} = \arg\min_{Z} \max_{\mu} \left\{ \alpha \cdot \mathbb{E}_{U(\tau)} \left[ \mathbb{E}_{\mathcal{D}(s),\mu(a|s)} F_{Z(s,a)}^{-1}(\tau) - \mathbb{E}_{\mathcal{D}(s,a)} F_{Z(s,a)}^{-1}(\tau) \right] + \mathcal{L}_p(Z, \hat{\mathcal{T}}^{\pi^k} \hat{Z}^k) \right\} + \mathcal{R}(\mu)$$

If we set $\mathcal{R}(\mu)$ to be the entropy $\mathcal{H}(\mu)$, then we can see that $\mu(a \mid s) \propto \exp(Q(s,a)) = \exp(\int_0^1 F_{Z(s,a)}^{-1}(\tau) d\tau)$ is the solution to the inner-maximization. Plugging this choice into the above regularized objective gives

$$\hat{Z}^{k+1} = \arg\min_{Z} \left\{ \alpha \cdot \mathbb{E}_{U(\tau)} \left[ \mathbb{E}_{\mathcal{D}(s)} \log \sum_{a} \exp(F_{Z(s,a)}^{-1}(\tau)) - \mathbb{E}_{\mathcal{D}(s,a)} F_{Z(s,a)}^{-1}(\tau) \right] + \mathcal{L}_p(Z, \hat{\mathcal{T}}^{\pi^k} \hat{Z}^k) \right\},$$

as desired. As in [21], we introduce a parameter $\zeta \in \mathbb{R}_{>0}$ that thresholds the quantile value difference between $\mu$ and $\hat{\pi}_\beta$. In addition, we scale this difference by $\omega \in \mathbb{R}_{>0}$. This gives a learnable formulation of $\alpha$ via dual gradient descent:

$$\min_{Z} \max_{\alpha \geq 0} \left\{ \alpha \cdot \mathbb{E}_{U(\tau)} \left[ \omega \cdot \left[ \mathbb{E}_{\mathcal{D}(s)} \log \sum_{a} \exp(F_{Z(s,a)}^{-1}(\tau)) - \mathbb{E}_{\mathcal{D}(s,a)} F_{Z(s,a)}^{-1}(\tau) \right] - \zeta \right] + \mathcal{L}_p(Z, \hat{\mathcal{T}}^{\pi^k} \hat{Z}^k) \right\},$$

Because our experiments are all conducted in continuous-control domains, we cannot enumerate all actions $a$ and compute $\log \sum_a \exp(F_{Z(s,a)}^{-1}(\tau))$ directly. To circumvent this issue, we use the importance sampling approximation scheme introduced in [21]. To this end, we use the following approximation in our implementation:

$$\log \sum_{a} \exp(F_{Z(s,a)}^{-1}(\tau)) \approx \log \left( \frac{1}{2M} \sum_{a_i \sim U(\mathcal{A})}^{N} \left[ \frac{\exp(F_{Z(s,a)}^{-1}(\tau))}{U(\mathcal{A})} \right] + \frac{1}{2M} \sum_{a_i \sim \pi(a|s)}^{N} \left[ \frac{\exp(F_{Z(s,a)}^{-1}(\tau))}{\pi(a_i \mid s)} \right] \right)$$

(15)

where $U(\mathcal{A}) = \text{Uniform}(\mathcal{A})$ denotes the uniform distribution over actions, and where we pick $M = 10$. We summarize a single step of the actor and critic updates used by CODAC in Algorithm 1.

## B.2 Neural network architecture

The policy network $\pi(\cdot \mid s; \phi)$ consists of a two-layer fully connected architecture with 256 hidden units and ReLU activations. For the quantile network, we use the architecture from [27], which builds on top of the implicit quantile network (IQN) architecture [6]. Specifically, we represent the quantile function $F_{Z(s,a)}^{-1}(\tau)$ as an element-wise (Hadamard) product of a state-action feature representation $\psi(s,a)$ and a quantile embedding $\varphi(\tau)$—i.e., $F_{Z(s,a)}^{-1}(\tau) = \psi(s,a) \odot \varphi(\tau)$. Following IQN, we use the following embedding formula for $\varphi(\tau)$:

$$\varphi_j(\tau) := h \left( \sum_{i=1}^{n} \cos(i\pi\tau) w_{ij} + b_j \right),$$

where $w_{ij}, b_j$ are weights of the neural network $\varphi$, and $h$ is the sigmoid function. We use a one-layer 256-unit fully connected neural network for $\psi(s,a)$, and a one-layer 64-unit fully connected neural network for $\varphi(\tau)$, followed with one-layer 256-unit fully connected network applied to $\psi(s,a) \odot \varphi(\tau)$. We apply layer normalization [2] after each activation layer to ensure stable training.

---

**Algorithm 1** CODAC Update

---

1: **Hyperparameters:** Number of generated quantiles $N$, quantile Huber loss threshold $\kappa$, CODAC penalty scale $\omega$, CODAC penalty threshold $\zeta$, discount rate $\gamma$, learning rates $\eta_{\text{actor}}, \eta_{\text{critic}}, \eta_\alpha$
2: **Parameters:** Critic parameters $\theta$, Actor parameters $\phi$, Penalty $\alpha$
3: **Inputs:** Tuple $s, a, r, s'$
4: Sample quantiles $\tau_i$ (for $i = 1, \ldots, N$) and $\tau'_j$ (for $j = 1, \ldots, N$) i.i.d. from $\text{Uniform}([0, 1])$
5:   # Compute distributional TD loss
6: Get next actions for calculating target $a' \sim \pi(\cdot \mid s'; \phi)$
7: **for** $i = 1$ **to** $N$ **do**
8:   **for** $j = 1$ **to** $N$ **do**
9:     $\delta_{\tau_i, \tau'_j} = r + \gamma F^{-1}_{Z(s',a'),\theta'}(\tau'_j) - F^{-1}_{Z(s,a),\theta}(\tau_i)$
10:   **end for**
11: **end for**
12: Compute $\mathcal{L}_{\text{critic}}(\theta) = N^{-2} \sum_{i=1}^{N} \sum_{j=1}^{N} \mathcal{L}_\kappa(\delta_{\tau_i, \tau'_j}; \tau_i)$
13:   # Compute CODAC penalty
14: Sample $i \sim U(\{1, ..., N\})$ and use quantile $\tau_i$
15: Estimate $\log \sum_a \exp(F^{-1}_{Z(s,a),\theta}(\tau_i))$ according to (15)
16: Compute $\mathcal{L}_{\text{CODAC}}(\theta, \alpha) = \alpha \cdot \left( \omega \cdot \left( \log \sum_a \exp(F^{-1}_{Z(s,a),\theta}(\tau_i)) - N^{-1} \sum_{j=1}^{N} F^{-1}_{Z(s,a),\theta}(\tau_j) \right) - \zeta \right)$
17: Update $\theta \leftarrow \theta - \eta_{\text{critic}} \nabla_\theta (\mathcal{L}_{\text{critic}}(\theta) + \mathcal{L}_{\text{CODAC}}(\theta, \alpha))$
18: Update $\alpha \leftarrow \alpha - \eta_\alpha \nabla_\alpha \mathcal{L}_{\text{CODAC}}(\theta, \alpha)$
19:   # Update Policy Network $\pi_\phi(a \mid s)$ with $\Phi_g$ objective
20: Get new actions with re-parameterized samples $\tilde{a} \sim \pi(\cdot \mid s; \phi)$
21: Compute $\Phi_g(s, \tilde{a})$ using $F^{-1}_{Z(s,\tilde{a}),\theta}(\tau_i), i = 1, ..., N$
22: $\mathcal{L}_{\text{actor}}(\phi) = \log(\pi(\tilde{a} \mid s; \phi)) - \Phi_g(s, \tilde{a})$
23: Update $\phi \leftarrow \phi + \eta_{\text{actor}} \nabla \mathcal{L}_{\text{actor}}(\phi)$

---

### B.3 Actor-Critic updates

We summarize a single actor-critic update performed by CODAC in Algorithm 1. We briefly discuss a few implementation details. First, since computing the CODAC penalty to all quantiles is prohibitively expensive, we apply the conservative penalty to a randomly chosen $\tau_i$ on each update step (Line 13-15). This practical choice aligns well with our theoretical objective, whose outer expectation is taken with respect to the uniform distribution $U(\tau)$ over quantiles. We also found subtracting the *average* quantile values (i.e., $N^{-1} \sum_{j=1}^{N} F^{-1}_{Z(s,a),\theta}(\tau_j)$) to be more stable than just subtracting the corresponding quantile value $F^{-1}_{Z(s,a),\theta}(\tau_i)$. This step can be viewed as rewriting

$$\mathbb{E}_{U(\tau)} \left[ \mathbb{E}_{\mathcal{D}(s)} \log \sum_a \exp(F^{-1}_{Z(s,a)}(\tau)) - \mathbb{E}_{\mathcal{D}(s,a)} F^{-1}_{Z(s,a)}(\tau) \right]$$

as

$$\mathbb{E}_{U(\tau)} \left[ \mathbb{E}_{\mathcal{D}(s)} \log \sum_a \exp(F^{-1}_{Z(s,a)}(\tau)) \right] - \mathbb{E}_{U(\tau)} \left[ \mathbb{E}_{\mathcal{D}(s,a)} F^{-1}_{Z(s,a)}(\tau) \right]$$

and implementing the latter as in Line 15. Finally, to compute $\Phi_g(s, \tilde{a})$ in Line 21, we take the average of all $F^{-1}_{Z(s,\tilde{a}),\theta}(\tau_i)$ where $\tau_i$ is less than or equal to the risk threshold value. For the expected-return (i.e., risk-neutral objective), the threshold is 1, and $\Phi_g(s, \tilde{a}) = \sum_{i=1}^{N} F^{-1}_{Z(s,\tilde{a}),\theta}(\tau_i)/N$. For CVaR0.1, the threshold is 0.1, and $\Phi_g(s, \tilde{a}) = \sum_{i=1}^{\max_i : \tau_i < 0.1} F^{-1}_{Z(s,\tilde{a}),\theta}(\tau_i)/(\max_i : \tau_i < 0.1)$.

## C  Experiment details and additional results

### C.1  Risky robot navigation

**Risky PointMass environment.** The state space of the PointMass agent 4-dimensional, including the agent's position as well as the goal position, which is fixed to $[0.1, 0.1]$. The state space constraint

Table 5: CODAC can optimize various distorted expectation based risk-sensitive objectives.

| Algorithm | Risky PointMass | | | |
| | Mean | Median | CVaR$_{0.1}$ | Violations |
|---|---|---|---|---|
| CODAC-CVaR | -6.05 | -4.89 | **-14.73** | **0.0** |
| CODAC-CPW | -8.34 | **-4.00** | -54.18 | 103.0 |
| CODAC-Neutral | -8.60 | -4.05 | -51.96 | 108.3 |
| CODAC-Wang | **-6.01** | -4.46 | -16.80 | 7.0 |

Table 6: Risky robot navigation quantitative evaluation.

| Algorithm | Risky PointMass | | | | Risky Ant | | | |
| | Mean | Median | CVaR$_{0.1}$ | Violations | Mean | Median | CVaR$_{0.1}$ | Violations |
|---|---|---|---|---|---|---|---|---|
| DSAC (Online) | -7.69 | -3.82 | -49.9 | 94 | -866.1 | -833.3 | -1422.7 | 2247 |
| CODAC-C (Ours) | **-6.05** $\pm$ 0.42 | -4.89 $\pm$ 0.35 | **-14.73** $\pm$ 0.95 | **0.0** $\pm$ 0.0 | -456.0 $\pm$ 24.0 | -433.4 $\pm$ 17.1 | **-686.6** $\pm$ 149.8 | **347.8** $\pm$ 69.7 |
| CODAC-N (Ours) | -8.60 $\pm$ 1.62 | **-4.05** $\pm$ 0.12 | -51.96 $\pm$ 12.34 | 108.3 $\pm$ 11.90 | **-432.7** $\pm$ 41.3 | **-395.1** $\pm$ 11.5 | -847.1 $\pm$ 309.3 | 936.0 $\pm$ 186.1 |
| ORAAC | -10.67 $\pm$ 1.18 | -4.55 $\pm$ 0.55 | -64.12 $\pm$ 5.14 | 138.7 $\pm$ 16.4 | -788.1 $\pm$ 82.0 | -795.3 $\pm$ 144.4 | -1247.2 $\pm$ 48.0 | 1196 $\pm$ 49.7 |
| CQL | -7.51 $\pm$ 1.05 | -4.18 $\pm$ 0.13 | -43.44 $\pm$ 10.57 | 93.4 $\pm$ 0.94 | -967.8 $\pm$ 66.9 | -858.5 $\pm$ 22.0 | -1887.3 $\pm$ 236.1 | 1854.3 $\pm$ 369.1 |

is $[0, 1]$. Hence, the agent cannot enter a location outside of this unit square. The risky red region is centered at $[0.5, 0.5]$ with radius of $0.3$. The agent's initial state is randomly chosen inside the $[0.1, 0.9]^2$ box outside the risky red region. The agent dynamics is holomorphic, allowing the agent to move freely in any direction with its $x$-axis and $y$-axis displacement capped at $0.1$. The reward the agent receives at each step is its negative Euclidean distance to the goal plus a constant $-0.1$, which encourages the agent to reach the goal as fast as possible. When the agent is inside the risky red region, with probability $0.1$, an additional $-50$ reward is incurred. The episode terminates when the agent is within $0.1$ distance to the goal. An episode may last up to $100$ steps.

**Risky Ant environment.** The state space of the Ant agent is identical to the original state space of the Mujoco Ant agent. The goal is located at $[10, 10]$, and the risky red region is centered at $[5, 5]$ with a radius of $3$. The agent's initial state is randomly chosen inside the $[0, 7]^2$ box outside the risky red region. The agent dynamics is also identical to the Mujoco Ant environment. At each timestep, the agent receives its negative Euclidean distance to the goal plus $0.1 \times$ velocity as its reward. When the agent is inside the risky red region, with probability $0.1$, an additional $-50$ reward is incurred. The episode terminates when the agent is within $0.1$ distance to the goal. When the agent is inside the risky red region, with probability $0.05$, an additional $-90$ reward is incurred. The episode terminates when the agent is within distance $1$ of the goal. An episode may last up to $200$ steps.

**Dataset and training details.** We train a distributional SAC agent online for $100$ (resp., $5000$) episodes in the PointMass (resp., Ant) environment, and use this agent's replay buffer as the dataset for offline RL training. All offline RL algorithms are trained for $10^4$ (resp., $10^6$) gradient steps. We use the default hyperparameters for ORAAC, and use $\omega = 0.01$ and $\zeta = 10$ for both CODAC and CQL. Our results are reported using $100$ evaluation episodes with same set of initial states.

**Additional results.** In Table 6, we show full results for the risky robot navigation environments. As can be seen, CODAC-C achieves the best performance on most metrics and is the only method that learns risk-averse behavior. In addition, in Figure 5, we visualize trajectories for various Ant agents. As can be seen, CODAC-C avoids the risky region shown in red, while still making it to the goal.

**Alternative risk-sensitive objectives.** On the risky pointmass domain, we also show that CODAC can optimize CPW and Wang risk-sensitive objectives using the same offline dataset. As for CODAC-CVaR (CODAC-C) and CODAC-Neutral (CODAC-N), we train CODAC-Wang and CODAC-CPW using 5 random seeds and report the results in Table 5. As shown, CODAC-Wang performs similarly to CODAC-CVaR, trading off slightly better average performance at the cost of safety. On the other hand, CODAC-CPW is on par with CODAC-Neutral. These findings match our intuition that Wang is slightly more risk-seeking than CVaR since it gives non-zero (but vanishingly small) weight to quantile values above the risk cutoff threshold, and CPW is similar to risk-neutral due to its intended modeling of human game-play behavior. These findings are also consistent with those in prior work [6], which investigates these risk objectives for online distributional reinforcement learning.

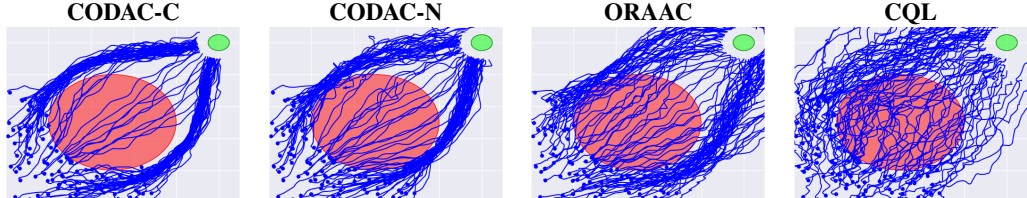

| CODAC-C | CODAC-N | ORAAC | CQL |
|---------|---------|-------|-----|

Figure 5: 2D visualization of evaluation trajectories on the Risky Ant environment. The red region is risky, the solid blue circles indicate initial states, and the blue lines are trajectories. CODAC-C learns the most risk-averse behavior while consistently approaching the goal.

Table 7: Normalized Return on the Stochastic D4RL Mujoco Suite, averaged over 5 random seeds.

|  | Algorithm | Medium | | Mixed | | Expert | |
|---|---|---|---|---|---|---|---|
|  |  | Mean | $\text{CVaR}_{0.1}$ | Mean | $\text{CVaR}_{0.1}$ | Mean | $\text{CVaR}_{0.1}$ |
| Cheetah | CQL | $33.2 \pm 21.6$ | $-15.0 \pm 14.3$ | $214.1 \pm 52.0$ | $12.0 \pm 23.8$ | $-74.8 \pm 22.6$ | $-206.6 \pm 46.9$ |
|  | ORAAC | $361.4 \pm 14.2$ | $91.3 \pm 42.1$ | $307.1 \pm 5.8$ | $118.9 \pm 27.1$ | $598.3 \pm 47.0$ | $99.7 \pm 71.3$ |
|  | CODAC-N | $338.9 \pm 65.7$ | $-41.6 \pm 16.7$ | $347.7 \pm 32.3$ | $149.2 \pm 79.2$ | $686.3 \pm 128.8$ | $123.2 \pm 90.1$ |
|  | CODAC-C | $335.8 \pm 80.6$ | $-27.7 \pm 60.3$ | $396.4 \pm 56.1$ | $238.5 \pm 58.9$ | $551.6 \pm 129.4$ | $151.3 \pm 133.0$ |
| Hopper | CQL | $877.9 \pm 193.3$ | $693.0 \pm 160.9$ | $189.2 \pm 63.0$ | $-21.4 \pm 62.5$ | $1165.0 \pm 59.4$ | $886.0 \pm 132.7$ |
|  | ORAAC | $1007.1 \pm 58.5$ | $767.6 \pm 101.0$ | $876.3 \pm 86.7$ | $524.9 \pm 323.0$ | $1156.8 \pm 340.5$ | $767.4 \pm 372.6$ |
|  | CODAC-N | $993.7 \pm 32.9$ | $952.5 \pm 29.0$ | $1483.9 \pm 16.2$ | $1457.6 \pm 20.7$ | $1292.7 \pm 34.9$ | $1024.0 \pm 45.6$ |
|  | CODAC-C | $1014.0 \pm 281.7$ | $976.4 \pm 272.1$ | $1551.2 \pm 33.4$ | $1449.6 \pm 101.3$ | $1270.6 \pm 74.8$ | $986.4 \pm 99.7$ |
| Walker2d | CQL | $1524.3 \pm 87.9$ | $1343.8 \pm 248.2$ | $74.3 \pm 76.7$ | $-64.0 \pm -77.7$ | $2045.2 \pm 37.6$ | $1868.2 \pm 55.1$ |
|  | ORAAC | $1134.1 \pm 235.4$ | $663.0 \pm 349.8$ | $222.0 \pm 37.4$ | $-69.6 \pm 76.3$ | $991.2 \pm 203.5$ | $108.9 \pm 73.2$ |
|  | CODAC-N | $1537.3 \pm 65.8$ | $1158.8 \pm 357.3$ | $358.7 \pm 125.4$ | $106.4 \pm 146.9$ | $2170.3 \pm 22.7$ | $2035.4 \pm 39.9$ |
|  | CODAC-C | $1120.8 \pm 319.3$ | $902.3 \pm 492.0$ | $450.0 \pm 193.2$ | $261.4 \pm 231.3$ | $2056.7 \pm 43.1$ | $1889.4 \pm 28.6$ |

## C.2 Stochastic D4RL Mujoco suite

Our experimental protocol largely follows [10]. All algorithms are trained for 500k gradient steps. We use 10 evaluation episodes on the modified Mujoco environments (see below). Hyperparameters are detailed in Appendix C.4.

**Dataset descriptions.** We describe the stochastic reward modification made to the original HalfCheetah, Hopper, and Walker2d environments [43]. These reward modifications are used to relabel the reward label in D4RL datasets; the modified environments are also used for evaluation in this set of experiments. The following paragraphs are adapted from [43]:

- **Half-Cheetah:** We use $R_t(s,a) = \bar{r}_t(s,a) - 70 \cdot \mathbb{1}_{v > \bar{v}} \cdot \mathcal{B}_{0.1}$, where $\bar{r}_t(s,a)$ is the original environment reward, $v$ is the forward velocity, and $\bar{v}$ is a threshold velocity ($\bar{v} = 4$ for Medium/Mixed datasets and $\bar{v} = 10$ for the Expert dataset). The maximum episode length is reduced to 200 steps.

- **Walker2D/Hopper:** We use $R_t(s,a) = \bar{r}_t(s,a) - p \cdot \mathbb{1}_{|\theta| > \bar{\theta}} \cdot \mathcal{B}_{0.1}$, where $\bar{r}_t(s,a)$ is the original environment reward, $\theta$ is the pitch angle, $\bar{\theta}$ is a threshold angle ($\bar{\theta} = 0.5$ for Walker2d and $\bar{\theta} = 0.1$ for Hopper) and $p = 30$ for Walker2d and $p = 50$ for Hopper. When $|\theta| > 2\bar{\theta}$ the robot falls, the episode terminates. The maximum episode length is reduced to 500 steps.

**Additional results.** In Table 7, we present the full Stochastic D4RL Mujoco results, including results on the Expert dataset. We repeat the results on the Medium and Mixed datasets in the main text here for completeness. Recall that the Expert (resp., Medium) dataset consists of rollouts from a fixed SAC agent trained to Expert (resp., Medium) performance, Expert is convergence and Medium is 50% of Expert performance. The Mixed dataset is the replay buffer of a SAC agent trained to achieve 50% of the expert return.

**Qualitative analysis.** To better interpret the stochastic D4RL results, we have collected behavioral statistics of the agents trained on the risk-sensitive HalfCheetah-Mixed-v0 and Walker2d-Mixed-v0 datasets. We execute one trained agent for each method reported in Table 2 for 10 episodes in the environment and record the percentage of timesteps where the agent violates the threshold and their average velocity over these evaluation episodes.

Table 8: Stochastic D4RL qualitative results

| Algorithm | HalfCheetah-Mixed-v0 | | Walker2d-Mixed-v0 | |
|---|---|---|---|---|
| | % Violation | Average Velocity | % Violation | Average Velocity |
| CODAC-C (Ours) | **11** | **1.49** | 15 | **0.28** |
| CODAC-N (Ours) | 54 | 2.02 | **9** | 0.34 |
| CQL | 23 | 1.71 | 13 | 0.19 |
| ORAAC | 37 | 1.76 | 48 | 0.49 |

Table 9: Normalized Return on the D4RL Mujoco Suite, averaged over 5 random seeds.

| Dataset | BC | BEAR | BRAC-v | BCQ | MOPO | CQL | ORAAC | CODAC |
|---|---|---|---|---|---|---|---|---|
| halfcheetah-random | 2.1 | 25.1 | 24.1 | 2.2 | **35.4** | 35.4 | 13.5 | 34.6 ± 1.27 |
| hopper-random | 9.8 | 11.4 | **12.2** | 10.6 | 11.7 | 10.8 | 9.8 | 11 ± 0.43 |
| walker2d-random | 1.6 | 7.3 | 1.9 | 4.9 | 13.6 | 7.0 | 3.2 | **18.7** ± 4.5 |
| halfcheetah-medium | 36.1 | 41.7 | 43.8 | 40.7 | 42.3 | 44.4 | 41.0 | **46.3**± 0.98 |
| walker2d-medium | 6.6 | 59.1 | 81.1 | 53.1 | 17.8 | 79.2 | 27.3 | **82.0** ± 0.45 |
| hopper-medium | 29.0 | 52.1 | 31.1 | 54.5 | 28.0 | 58.0 | 1.48 | **70.8** ± 11.4 |
| halfcheetah-mixed | 38.4 | 38.6 | 47.7 | 38.2 | **53.1** | 46.2 | 30.0 | 44 ± 0.76 |
| hopper-mixed | 11.8 | 33.7 | 0.6 | 33.1 | 67.5 | 48.6 | 16.3 | **100.2** ± 1.0 |
| walker2d-mixed | 11.3 | 19.2 | 0.9 | 15.0 | **39.0** | 26.7 | 28 | 33.2 ± 17.6 |
| halfcheetah-medium-expert | 35.8 | 53.4 | 41.9 | 64.7 | 63.3 | 62.4 | 24.0 | **70.4** ± 19.4 |
| walker2d-medium-expert | 6.4 | 40.1 | 81.6 | 57.5 | 44.6 | 98.7 | 28.2 | **106.0** ± 4.6 |
| hopper-medium-expert | 111.9 | 96.3 | 0.8 | 110.9 | 23.7 | 111.0 | 18.2 | **112.0** ± 1.7 |

As shown in Table 8, CODAC-C achieves the lowest percentage of violations in the HalfCheetah environment, indicating that it has learned a safer policy than all other methods. On Walker2d, CQL appears to be the safest; however, this result is due to the fact that CQL failed to learn the desirable walking behavior as indicated by its low reward in the paper. Among the methods that learned to walk, CODAC-C achieves the lowest average angular velocity while maximizing the return.

### C.3 D4RL Mujoco suite

Our experimental protocol largely follows from [10]. All algorithms are trained for 1M gradient steps. We use 10 evaluation episodes on the original Mujoco environments, which all last 1000 steps long. Hyperparameters are detailed in Appendix C.4. In Table 9, we show the full results on the risk-neutral D4RL Mujoco Suite, which includes additional baselines such as BEAR [20] and BRAC [46].

### C.4 Hyperparameters

As CODAC builds on top of distributional SAC (DSAC), we keep the DSAC-specific hyperparameters identical as the original work. These hyperparameters are shown in Table 10.

CODAC additionally introduces hyperparameters $\alpha, \omega, \zeta$ (see Appendix B). In most cases, $\alpha$ is a learnable parameter initialized to 1 with learning rate $\eta_\alpha = 3 \times 10^{-4}$; in few cases, we fix it to 1 throughout the entirety of training, which we indicate by setting $\zeta = -1$, as in [21]. For ORAAC, we use the default hyperparameters tuned on the stochastic D4RL Mujoco suite for all experiments; for CQL, we use the default hyperparameters tuned on the original D4RL Mujoco suite for all experiments. Below, we describe the specific CODAC hyperparameters we use for the risk-neutral and risk-sensitive D4RL experiments.

**Risk-neutral D4RL.** We restrict the search range of the hyperparameters as follow: $\omega \in \{0.1, 1, 10\}, \zeta \in \{-1, 10\}$. We also experiment with enabling entropy tuning in DSAC and tune the value network learning rate $\eta_{\text{critic}}$ between $3e-4$ and $3e-5$, which improves performance on some datasets. Table 11 summarizes the hyperparameters used for each dataset in our reported results. At a high level, we find $\omega = 1$ to be effective for Mixed and Random datasets and $\omega = 10$ effective for Medium and Medium-Expert datasets. These empirical findings match our intuition that the penalty needs not to be high when the dataset has wide coverage.

Table 10: CODAC backbone hyperparameters

| Hyper-parameter | Value |
|---|---|
| Policy network learning rate $\eta_{\text{actor}}$ | 3e-4 |
| (Quantile) Value network learning rate $\eta_{\text{critic}}$ | 3e-5 |
| Optimizer | Adam |
| Discount factor $\gamma$ | 0.99 |
| Target smoothing | 5e-3 |
| Batch size | 256 |
| Replay buffer size | 1e6 |
| Minimum steps before training | 1e4 |
| Number of quantile fractions $N$ | 32 |
| Quantile fraction embedding size | 64 |
| Huber regression threshold $\kappa$ | 1 |

Table 11: CODAC hyperparameters for risk-neutral D4RL

| **dataset** | $\omega$ | $\zeta$ | $\eta_{\text{critic}}$ | entropy tuning |
|---|---|---|---|---|
| halfcheetah-random | 1 | 10 | 3e-5 | yes |
| hopper-random | 1 | 10 | 3e-5 | yes |
| walker2d-random | 1 | 10 | 3e-5 | yes |
| halfcheetah-medium | 10 | 10 | 3e-5 | no |
| hopper-medium | 10 | 10 | 3e-4 | yes |
| walker2d-medium | 10 | 10 | 3e-5 | no |
| halfcheetah-mixed | 1 | 10 | 3e-5 | yes |
| hopper-mixed | 1 | 10 | 3e-5 | yes |
| walker2d-mixed | 1 | 10 | 3e-5 | yes |
| halfcheetah-medium-expert | 0.1 | -1 | 3e-4 | no |
| hopper-medium-expert | 10 | 10 | 3e-5 | no |
| walker2d-medium-expert | 10 | 10 | 3e-5 | no |

**Risk-sensitive D4RL.** We use the same hyperparameter range as in risk-neutral D4RL for a grid search. Interestingly, the best value of $\omega$ is smaller across most datasets, suggesting less conservatism may be needed due to the increased stochasticity in the environment. Table 12 summarizes the hyperparameter choices.

## C.5 Compute resources

We use a single Nvidia 2080-Ti with 32 cores to run our experiments. Each CODAC run takes about 10 hours in clock time.

Table 12: CODAC hyperparameters for risk-sensitive D4RL

| dataset | $\omega$ | $\zeta$ | $\eta_{\text{critic}}$ | entropy tuning |
|---|---|---|---|---|
| halfcheetah-medium | 1 | -1 | 3e-5 | no |
| hopper-medium | 0.1 | 10 | 3e-5 | yes |
| walker2d-medium | 1 | -1 | 3e-5 | yes |
| halfcheetah-mixed | 0.1 | 10 | 3e-5 | yes |
| hopper-mixed | 1 | 10 | 3e-5 | yes |
| walker2d-mixed | 1 | 10 | 3e-5 | yes |
| halfcheetah-medium-expert | 1 | -1 | 3e-5 | yes |
| hopper-medium-expert | 10 | 10 | 3e-5 | no |
| walker2d-medium-expert | 10 | 10 | 3e-5 | no |