# OpenReview forum: "Conservative Offline Distributional Reinforcement Learning"
_NeurIPS.cc/2021/Conference — NeurIPS 2021 Poster_

### Official Review · Reviewer_9EDx · 2021-07-12

**Rating:** 6
**Confidence:** 4

**Summary:**

This paper adapts distributional RL to the offline setting by penalizing the predicted quantiles of the return for out-of-distribution actions. It proposes a CODAC algorithm for this purpose, proves the convergence of the proposed alg in tabular cases, and shows that it can learn risk-averse policies from offline data collected purely from risk-neutral policies.

**Ethical Concerns:**

No concern.

**Limitations And Societal Impact:**

The authors have adequately addressed the limitations and social impact of their work.

**Main Review:**

**Strengths**:
-	The use of distributional RL for offline learning is a promising direction and this paper explores just that. I think such a combination makes an important bit to further improve the performance of offline RL which is becoming a big deal for RL in practical settings.
-	The experimental results on risk-sensitive and risk-neutral policies are promising though expected
**Weaknesses**:
-	The approach proposed is quite simple and straightforward without much technical innovation. For example, CODAC is a direct combination of CQL and QR-DQN to learn conservative quantiles of the return distribution.
-	Some parts of the paper need clearer writing  (more below)

**Comments and questions**:
-	I think in a paragraph from lines 22-30 when discussing distributional RL, the paper lacks relevant literature on using moment matching (instead of quantile regression as most DRL methods) for DRL (Nguyen-Tang et al AAAI’21, “Distributional Reinforcement Learning via Moment Matching”). I think this should be properly discussed when talking about various approaches to DRL that have been developed so far, even though the present paper still uses quantile regression instead of moment matching.
-	More explanation is needed for Eq (5). For example, what is the meaning of the cost $c_0(s,a)$? (e.g., to quantify out-of-distribution actions)
-	The use of $s’$ and $a’$ when defining $\hat{\pi}_{\beta}$ at line 107 might cause confusion as $\mathcal{D}$ contains $(s,a,r,s’)$.
-	This paper is about deriving a conservative estimate of the quantiles of the return from offline data where the conservativeness is for penalizing out-of-distribution actions. In the paper, they define OOD actions as those are not drawn from \hat{\pi}_{\beta}(.|s) (line 109) but in Assumption 3.1. they assume that \hat{\pi}_{\beta}(a|s) > 0, i.e., there is no OOD actions. Thus, what is the merit of the theoretical result presented in the paper?

**Time Spent Reviewing:**

5

---

> ### Author Response · Authors · 2021-08-10
> **Response to Reviewer 9EDx**
>
> We thank the reviewer for their constructive feedback. Besides the response below, we will include (1) the provided reference (Nguyen-Tang et al, AAAI'21), (2) discussion on distributional-matching based distributional RL algorithms in related work, and (3) corrections on notations in our next paper draft. Now, we respond to some conceptual questions the reviewer raises. Please let us know if we can provide any additional clarifications during the discussion period.
>
> ---
> **Comment 1**: The proposed approach is simple and without technical innovation; CODAC is a combination of CQL and QR-DQN.
>
> **Response 1**: We believe that characterizing CODAC as a combination of CQL and QR-DQN is incorrect, and neglects our work's algorithmic and theoretical contributions. We will greatly clarify our contributions in our next paper revision. Here, we summarize them at a high level.
>
> On the algorithmic side, a key contribution is our objective Equation 5, which extends CQL to the distributional setting. It differs from CQL both in that (i) the penalty in the first term is applied to the quantiles $F_{Z(s,a)}^{-1}(\tau)$ rather than the $Q$-values, and (ii) the second term is in terms of the $p$-Wasserstein distance instead of the mean-square error. A priori, it is not obvious how to do so in a way that provides guarantees on the quantile function (or even that lower bounding the quantile function is a reasonable strategy).
>
> On the theoretical side, we provide significant theoretical analysis justifying Equation 5. In particular, we lay out the necessary conditions for the lower bound to hold. Proving that Bellman iteration using Equation 5 converges (Lemma 3.4, 3.5), that it obtains uniform conservative quantile lower bound (Theorem 3.6), and that it satisfies the gap-expansion property (Theorem 3.8), all require novel proof techniques. For example, they include (1) the use of calculus of variations (Appendix A.1), (2) a concentration bound on the quantile function (Lemma A.4, A.5), and (3) novel properties of the distributional Bellman operator (Lemma A.2, A.6). Due to space limits, we relegate these results to the Appendix, highlighting only the core theorems in the main text along with their direct consequences.
>
> Together, we view the fact that CODAC ultimately appears to be a natural extension of CQL to distributional RL as a key benefit of our approach; its naturalness and simplicity is a direct result of our careful design choices and theoretical analysis.
>
> Finally, we note that QR-DQN parameterizes policy return distribution using a finite set of discrete quantiles; this coarse approximation does not permit flexible computation of various risk objectives (e.g., CVaR with different risk thresholds). Therefore, we in fact use implicit quantile network (IQN), an extension of QR-DQN which admits value for all quantiles, as our backbone distributional RL algorithm. We note that justifying this architectural choice is achieved through synthesizing our contributions summarized above.
>
> ---
> **Comment 2**: Does the definition of OOD action (Line 109) and Assumption 3.1 contradict each other?
>
> **Response 2**: We consider an action (to be precise, a state-action pair $(s,a)$) to be OOD if it has low density under the behavior policy $\pi_\beta$ but high probability under the learned policy $\pi$, i.e., $\pi(a\mid s)\gg\pi_\beta(a\mid s)$. This definition does not preclude an OOD action from having non-zero probability under the behavior policy distribution. Thus, Assumption 3.1 does not contradict the presence of OOD actions according to our definition. Note that our definition is consistent with those in CQL as well as a recent offline RL tutorial [A].
>
> This definition of OOD is particularly relevant in offline RL; the $Q$-value estimate of an OOD action intrinsically has high variance (if $\pi_\beta(a\mid s)$ is small, even if it is $>0$) and can be easily exploited by the learned policy $\pi$ if its uncertainty is not accounted for. We acknowledge that OOD has different usages across machine learning, and we will clarify our intended definition in our next paper draft.
>
> With respect to the merit of our theoretical results, we note that this assumption is only needed to avoid division-by-zero in our bounds. In general, it is impossible to provide theoretical bounds if there are state-action pairs with zero observations (without further assumptions on the MDP), since we cannot estimate the transitions and rewards at that pair. This assumption has no bearing on our practical algorithm CODAC, evidenced by the fact that our experiments are conducted on continuous control domains.
>
> [A] Levine, Sergey, et al. "Offline reinforcement learning: Tutorial, review, and perspectives on open problems." arXiv preprint arXiv:2005.01643 (2020).
>
> ---
> **Comment 3**: Equation 5 needs clearer explanation. For example, what is $c_0(s,a)$?
>
> **Response 3**: Equation 5 is a combination of the $p$-Wasserstein distributional RL policy evaluation objective (the $\mathcal{L}_p$ term) and the CODAC penalty term (the expectation term including $c_0$). Together, they give rise to an objective that can be optimized offline and converges to a fixed point (Lemma 3.5) that lower bounds the true quantile values of the policy (Theorem 3.6).
>
> In particular, $c_0(s,a)$ is a scaling factor that determines the conservative penalty applied to that particular $(s,a)$ in the distributional critic update. For Theorem 3.6 to hold, it can be any arbitrary user-provided function satisfying $c_0(s,a)>0$. We provide a particular choice in Equation 6, which is required for Theorem 3.8 to hold. Intuitively, in this choice, $c_0(s,a)$ is larger for those actions that are infrequent with respect to the behavior policy. We will provide a more detailed discussion on $c_0(s,a)$ as well as Equation 5 in our next paper draft.

---

> > ### Comment · Reviewer_9EDx · 2021-08-24
> > **Response to the author's comment**
> >
> > Thank you for the detailed rebuttal. I have read the author's response and the other reviews. I am satisfied with the author's response and happy to keep my initial score. I encourage the authors to include the comments and discussion here in the paper.

---

### Official Review · Reviewer_Haog · 2021-07-15

**Rating:** 7
**Confidence:** 3

**Summary:**

The paper introduces a general purpose offline distributional reinforcement learning algorithm CODAC. They prove that CODAC obtains conservative estimates of the return quantile, which translate into lower bounds on values. They provide sound theoretical and thorough empirical analysis for the proposed method.

**Limitations And Societal Impact:**

Limitations and impacts are addressed.

**Main Review:**

The paper is well organized and well written in general. The results presented in the paper looks interesting and has potential of bringing good impact in risk-averse RL.

What is x in equation (2). Is there an intuitive way to interpret what's happening here?

The quantile based formulation is varsatile and is a good addition for risk based optimization. One can get VaR, CVaR and so on simply by updating the quantile. CVaR is known to be not time-consistent. Or in other words, a recursive decomposition does not hold and the policy is history dependent for CVaR optimization in MDPs, unless the state space is augmented [1]. But this paper is providing a recursive formulation without augmenting the state space, even for CVaR. An analytical study in this matter would be greatly appreciated.

The theoretical results presented in the paper is rigorous and looks good from a high level. Though I did not get the chance to verify the correctness of all the proofs presented in the paper.

In description of figure 1, the inital states are not represented by 'blue' color, as stated. May be green?

The empirical study presented in the paper is detailed and well thought. It does provide a comprehensive overview on the behavior of CODAC compared to other baseline methods.


[1] Risk-Sensitive and Robust Decision-Making: a CVaR Optimization Approach. Yinlam Chow, Aviv Tamar, Shie Mannor, Marco Pavone.

**Time Spent Reviewing:**

8

---

> ### Author Response · Authors · 2021-08-10
> **Response to Reviewer Haog**
>
> We thank the reviewer for their feedback and encouraging assessment of our paper's potential impact. Here, we respond to some clarifying questions the reviewer raises. Please let us know if we can provide any additional clarifications during the discussion period.
>
> ---
> **Comment 1**: An intuitive interpretation of Equation 2.
>
> **Response 1**: Equation 2 can be viewed as the analog of the one-step Bellman update for the distributional RL setting; in particular, it is the Bellman update for the CDF functions $F_{Z(s,a)}$ of the discounted return distribution $Z(s,a)$ (which is a random variable) for each state-action pair $(s,a)$. Intuitively, $F_{Z(s,a)}(x)$ says: "*Considering all future randomness, where does $x$ land in the distribution of all possible future discounted return?*''
>
> In more detail, $x$ represents a particular value in the support of the discounted return distribution. Then, the left-hand-side of Equation 2 denotes the percentile of $x$ in this distribution after taking one step in the environment from the current state and action $(s,a)$. Because a step is taken in the environment according to the policy, this quantity needs to be dependent on the policy-conditioned transition probability of the next state-action pair $(s',a')$ from the current $(s,a)$. Then, given a particular $(s',a')$, where $x$ lands on the return distribution depends on (1) the distribution of reward the agent obtains at $(s,a)$, and (2) the distribution of discounted return obtained at $(s',a')$ and onward. The latter point leads to the recursive nature of Equation 2.
>
> Now, given that $x$ is obtained as the total discounted return, for any reward $r$ that were obtained from distribution $R(s,a)$ at the current timestep, the policy must "accrue'' $(x-r)/\gamma$ discounted return starting at $(s',a')$. Because both the reward function R and $F_{Z(s',a')}$ are random variables, their dependency is captured precisely by their convolution (i.e., the integral in the right-hand-side). Putting everything together, we have the equality in Equation 2.
>
> Finally, for a rigorous derivation, we refer the reader to [A], in particular, Equations (20) to (26) in their appendix.
>
> ---
> **Comment 2**: How does CODAC's use of distributional Bellman operator (i.e., recursive decomposition) reconcile with the time-inconsistency of CVaR objective?
>
> **Response 2**: We emphasize that our theoretical results pertain to offline distributional policy *evaluation*, instead of optimization. Therefore, our recursive application of distributional Bellman evaluation operator is valid and will converge to a fixed point. Note that this recursive decomposition is based on one used by a prior work [A] (see Equations 20 to 26). In particular, this recursive decomposition in itself does not depend on CVaR, which is a particular choice of policy objective we optimize. Because CVaR in our setting is a continuous integration of the quantiles, we are able to provide a conservative lower bound guarantee on offline CVaR policy evaluation in the tabular MDP case. CODAC leverages this policy evaluation algorithm in an actor-critic framework to perform policy optimization; here, the actor optimization is based on existing heuristics for CVaR policy optimizing [B].
>
> Furthermore, concurring with the reviewer, we emphasize that we do not claim convergence to the CVaR-optimal policy. We will make this distinction clearer in our next paper draft. Nevertheless, as our theoretical framework applies generally to any state space of an MDP, we can always augment the state space with the history information needed for CVaR-optimality (in particular, the reward accrued so far), which may lead to better empirical CVaR performance by resolving the stated time-inconsistency (though global optimality is still not guaranteed due to our use of a heuristic actor-critic algorithm).
>
> As an example, on the Risky PointMass environment, we train CODAC-C-Aug, the CVaR variant of CODAC trained on an augmented state space that includes the current return as a state. We divide the reward by 100 to ensure similar magnitude for the current return state variable and original state variables in the environment. Fixing the dataset and the hyperparameters, we compare CODAC-C-Aug and CODAC-C using 5 random seeds, and report the results in the table below. As shown, CODAC-C-Aug offers slightly improved CVaR performance, suggesting that state-augmentation may offer some advantage, though the improvement is not significant; this is expected as CODAC-C already achieves behaviorally optimal performance in the paper.
>
> |      Risky PointMass (State-Aug)       |     Mean | CVaR-0.1 |   |   |
> |-------------|----------:|------------|---|---|
> | CODAC-C     | -0.16     | -0.83      |   |   |
> | CODAC-C-Aug | **-0.14** | **-0.77**  |   |   |
> |             |           |            |   |   |
>
> [A] Keramati, Ramtin, et al. "Being optimistic to be conservative: Quickly learning a cvar policy." Proceedings of the AAAI Conference on Artificial Intelligence. Vol. 34. No. 04. 2020.
>
> [B] Ma, Xiaoteng, et al. "DSAC: Distributional Soft Actor Critic for Risk-Sensitive Reinforcement Learning." arXiv preprint arXiv:2004.14547 (2020).
>
> ---
> **Comment 3**: What do the small blue circles and the green circle represent in Figure 1?
>
> **Response 3**: The initial states are represented by the small blue circles; the bigger green circle is the fixed goal of the environment. Figure 1 shows that CODAC-C is able to navigate to the goal safely from a wide distribution of initial states without traversing through the risky red circle in the middle.

---

### Official Review · Reviewer_1we9 · 2021-07-15

**Rating:** 7
**Confidence:** 4

**Summary:**

At its core, the paper introduces an algorithm called conservative distributional evaluation (CDE). In the tabular settings, the CDE algorithms theoretically guarantees to converge to a fixed point whose quantile function lower bounds that of the true return distribution, also its fixed point is more conservative for rare actions in the dataset (the property that is called gap-expanding). These are desired properties for offline scenarios with full coverage (see the comments bellow). CDE is then used in an actor-critic framework, which is called Conservative Offline Distributional Actor Critic (CODAC). This gets a min-max objective: the inner loop chooses the current policy to maximize the CDE objective, and the outer loop minimizes the CDE objective for this policy. Some experimental results are followed to support the theory.

**Limitations And Societal Impact:**

This is a mostly theoretical work. Not at a level to directly cause societal impact.

**Main Review:**

-- The **far too strong** assumption that the dataset $\mathcal{D}$ has full coverage is ignored to discuss (Assumption 3.1). In real-world offline settings (such problem as in healthcare) we almost always deal with situations where only a tiny fraction of the state-action space is covered by $\mathcal{D}$. When $\mathcal{D}$ has full coverage, the problem is simply an **off-policy** learning problem (rather than offline) with nearly complete exploration. This is identical to the *online* case where there is an unknown behavioural policy for each experienced trajectory and where all state-action pairs are seen frequently enough. I found this case very misleading to be called **offline**.

-- The penalty term in (5) is sampled-based. In particular, $F^{-1}_{Z}(\tau)$ with uniform $\tau$ may simply get a large value by some chance; hence, incurring a large penalty even when $Z$ is not distributed close to uniform. This may cause an unnecessary instability. Why not using other measures of uncertainty which are deterministic function of distribution’s shape, such as moments of $Z$ or entropy (as entropy is a super-exponential lower-bound for all even moments). Such measures can be computed directly from the quantile values; the regularization term in (5) will then only contain an expectation over $\mathcal{D}$.

-- Proof of Lemma 3.4 (in the Appendix) -> the first term in expressions after line 518 -> expectation over $\tau$ should introduce an extra constant coming from the uniform distribution when writing the integral over $\tau$, which is missing. Further, $\alpha$ also is missing in the second line. (They shouldn’t change the result when the derivative is set to zero.)

-- Theorem 3.6 -> left-hand-side of the inequality must also be $F^{-1}$.

-- Theorem 3.6 -> A return distribution must sum to one, regardless. Intuitively, if the learned distribution lower bounds the original distribution in all the quantiles it cannot sum to one (unless at least in one (discrete) quantile it overestimates with significant amount such that it compensates for the rest). This theorem sounds counter-intuitive. Am I missing something? (This needs clarification in the text as well).

-- After equation (6) -> It is not clear where $\mu$ comes from and how sensitive the results are to this policy. A discussion here is needed.


Other comments:

-- Line 100: In general, $[V_{min}, V_{max}] \subseteq [\frac{R_{min}}{1-\gamma}, \frac{R_{max}}{1-\gamma}]$ instead of being equal. For instance, in a goal-seeking MDP, $[V_{min}, V_{max}] = [R_{min}, R_{max}]$.

-- Line 103 -> what do you mean by “its correspondence to policy”? If it refers to the optimal polic/value, then fix the last expression by adding asterisk.

-- Line 107 -> definition of $\hat{\pi}_{\beta}$: right hand side should be conditioned on (s,a).

-- Equation (5) -> what is $U$? this should be uniform distribution (?), yet in the previous section you used Uniform(.). If yes, it is better to be consistent.

-- Careful proofread is required. I saw at least a couple of sentences with no full-stop at the end.

**Time Spent Reviewing:**

7

---

> ### Author Response · Authors · 2021-08-10
> **Response to Reviewer 1we9**
>
> We thank the reviewer for their thoughtful feedback and careful proofreading of our manuscript; we will improve the exposition of our paper based on the provided feedback. Below, we address the reviewer's conceptual concerns. Please let us know if we can provide any additional clarifications during the discussion period.
>
> ---
> **Comment 1**: Assumption that the dataset has full coverage is too strong and makes the problem off-policy rather than offline.
>
> **Response 1**: This data coverage assumption is only needed in our theoretical analysis to avoid division-by-zero in our bounds, and is made by a prior work in offline RL that leverages similar penalty mechanism [A]. In general, it is impossible to obtain any optimality/lower bound guarantees for a state-action pair that has zero coverage in the dataset (without further assumptions on the MDP), since we would be unable estimate the transitions and rewards for that pair.
>
> Furthermore, our practical algorithm does not assume it or depend on it to obtain good performance. In practice, as suggested by the reviewer, the stated assumption is hardly ever satisfied due to either inherent data scarcity or continuous domains (e.g. robotics). It is certainly not satisfied in our experiments, which command continuous action spaces. Therefore, we believe that our work still falls under the umbrella of offline RL, and we will clarify the scope of this assumption in our next draft.
>
> Finally, our usage of ``offline RL'' is consistent with those in a recent tutorial [B], in which offline RL is defined as learning a policy from a fixed dataset without further environmental interaction (Section 2.2, bottom of Page 7); this definition of offline RL does not make explicit assumption on the data coverage or lack thereof. Nevertheless, we acknowledge that our current presentation of offline RL and the scope of our theoretical assumptions is likely a source of confusion, and will greatly improve their clarity in the next paper draft.
>
> [A] Kumar, Aviral, et al. "Conservative q-learning for offline reinforcement learning." arXiv preprint arXiv:2006.04779 (2020).
>
> [B] Levine, Sergey, et al. "Offline reinforcement learning: Tutorial, review, and perspectives on open problems." arXiv preprint arXiv:2005.01643 (2020).
>
> ---
> **Comment 2**: The penalty term in Equation 5 is sample-based and may incur instability.
>
> **Response 2**: In practice, we note that using sampling to estimate our objective will not result in instability as long as the support of $Z$ is bounded, which is typically true in reinforcement learning (in particular, we assume $Z\in[V_{\text{min}},V_{\text{max}}]$).
> Nevertheless, we note that our objective *is* a deterministic function of the shape of the distribution $F_{Z(s,a)}$ (in particular, it is the expected value of $F_{Z(s,a)}^{-1}(\tau)$ over $\tau\sim\text{Uniform}([0,1])$ and $(s,a)\sim\mathcal{D}$). Thus, we can in principle use algorithms such as quadrature in place of our sampling strategy without changing our objective. We found sampling to be an effective strategy for our benchmarks.
>
> ---
> **Comment 3**: Alternative choices of penalty mechanism may be preferrable in Equation 5.
>
> **Response 3**: Our penalty term, in particular taking $\tau$ to be uniform on $[0,1]$, is necessary for our theoretical guarantees. This choice enable us to lower bound every quantile value uniformly (Theorem 3.6) and to attain a lower bound for all quantile-based objectives (Corollary 3.7). With alternative penalty mechanisms such as the ones suggested by the reviewer, the uniform lower bound on quantiles is no longer guaranteed (they might result in alternative theoretical guarantees, which merit study in future work), and consequently, we cannot guarantee that offline optimization of quantile-based objectives can achieve good performance. Finally, even in the case that $Z$ has unbounded support (especially if it is heavy-tailed), we can use alternative integration algorithms to evaluate our objective, such as the quadrature strategy mentioned above.
>
>
> ---
> **Comment 4**: Clarification on return distribution in the statement of Theorem 3.6.
>
> **Response 4**: The left-hand-side of the inequality is indeed $F^{-1}$; we thank the reviewer for pointing out this typo. In general, the lower bound in Theorem 3.6 is on the quantiles of the distribution, which do not need to sum/integrate to one (only the density necessarily integrates to one). Intuitively, consider the CDF of the return distribution; then, Theorem 3.6 says that the estimate of the CDF is shifted to the left compared to the true CDF.
>
> As a concrete example, suppose that the return distribution is uniform on $[0,1]$, and we apply a constant penalty of $0.5$ to all quantiles. This penalty corresponds to horizontally shifting the quantile values (or equivalently, the CDF) to the left by $0.5$, which results in the uniform distribution on $[-0.5, 0.5]$. The quantile values of the latter distribution uniformly lower bound the corresponding quantile values of the original distribution.
>
> ---
> **Comment 5**: Where does $\mu$ come from in Equation 6?
>
> **Response 5**: Here, $\mu$ is any action distribution that is different from the behavior policy; we will define $\mu$ more clearly in the paper. Our gap-expansion result (Theorem 3.8) holds for any such $\mu$ given the choice of $c_0$ in Equation 6, so at a theoretical level, our result is not sensitive to the choice of $\mu$ as the gap-expansion result holds regardless. In CODAC, $\mu$ is heuristically chosen as the policy derived from the current iterate of the min-max policy objective (Equation 7). Since this choice of $\mu$ approximately maximizes the inner-loop of Equation 7, the fact that CODAC performs well empirically suggests that it is not sensitive to the choice of $\mu$.

---

> > ### Comment · Reviewer_1we9 · 2021-08-16
> > **After Rebuttal**
> >
> > I appreciate authors' feedback and clarifications. As my comments were largely aimed for improvement rather than raising concerns, my score remains the same, and I would vote for acceptance. Nevertheless, I would like to write down two remarks.
> >
> > 1) As for the loss function, my understanding was that the loss is calculated by uniformly sampling $\tau$ at the run time. Hence, even if the loss has an expectation operator mathematically, the actual computation is not deterministic and is dependent on the chosen random seed, mini-batch size, etc. My suggestions (not necessarily good ones) solely involve sampling on $\mathcal{D}$ and not $\tau$. I might have been wrong though. I don't see clarification in authors' response.
> >
> > 2) As for the common misuse of the term "offline", the response is not to the point (neither adding the [too recent] citations). I'd like to reiterate my core comment about the distinction between off-policy and offline. If there is full coverage, why should we care calling the problem offline?

---

### Official Review · Reviewer_8SBh · 2021-07-19

**Rating:** 7
**Confidence:** 4

**Summary:**

This paper tackles the offline RL setting, in which a policy is learned entirely from historical data, rather than through real-time online interaction with a simulator. However, existing algorithms such as CQL, while able to learn conservative lower bounds on the true Q-values, learn point estimates, making it difficult to also account for risk. Key contributions include:
1. bridging the gap between distributional RL and conservative value-based RL (CQL) for the first time, in which a lower bound on the quantiles of the return distribution can be learned through a penalization approach (CODAC).
2. two key theoretical results ensure that the method works as expected. Theorem 3.6 shows that CODAC will learn quantile estimates that lower bound the quantiles of the true return distribution. Intuitively, a lower bound is preserved when integrating over this distribution, such as when computing CVaR objectives. Theorem 3.8 shows gap-expansion, in which the degree of conservativeness is less for in-distributions actions than it is for out-of-distribution actions.
3. A practical actor-critic algorithm is developed based on the min-max formulation of the original penalized objective in CQL, but in the distributional RL setting. This algorithm is evaluated on a variety of robotics tasks, to see whether CODAC can learn conservative quantiles while being able to take risk into account.

**Limitations And Societal Impact:**

tightness of lower bounds: original work on CQL shows that the learned bounds are reasonably tight lower bounds on the Q-values. does a similar result hold for the quantiles as well?

visualization and interpretation of results: the results are quite hard to interpreted for the stochastic domains from tables alone. it is also hard to understand what kind of behavior the agent is learning on the stochastic domains. is it possible to visualize the learned behaviors similar to the first two domains?

failure modes: it is also hard to understand why the algorithm fails (e.g. risk sensitive cheetah task?). can any commentary be provided to discuss why the algorithm fails here and what could be done in the future to address this?

choice of risk measures: why is CVaR the approach used here, given it is not novel? it would be nice to provide further reasoning. existing work on distributional RL often allows other risk measures to be defined through other transforms of tau (e.g. Wang transform, etc.) Can the current work be applied for other such risk measures? Are most/all well studied risk measures quantile-based or are some choices not amenable to the analysis in this paper? it doesn't seem that exponential utilities or certainty-equivalents are possible for example, but I may be wrong.

overall: I believe this is a strong paper with non-trivial contributions. the overall approach makes sense and the theoretical results are quite intuitive. furthermore, the limitations of the approach (tuning alpha, etc) are anticipated but they are discussed in the conclusion, where it is noted that such limitations are not exclusive to this work, so I do not see this as an issue.



**Main Review:**

originality: I believe the paper has sufficient novelty for NeurIPS, being a novel combination of distributional RL and conservative Q-learning. This novel contribution also solves an important problem of learning risk-sensitive policies from offline data, which is a complex but important problem. existing methods are well discussed, and it is clear how this method differs in problem setting from CQL and work on distributional imitation learning.

clarity: overall, the paper is very well written and the motivation for studying this problem is well argued. some minor details below:

1. line 76 "paratemerizing" --> "parameterizing"
2. the notation in section 2, from line 92 is very dense, and some notation isn't well defined: is D(s, a) the data set restricted to s, a pairs?
3. it could really help readers to provide a simple motivating example early on (before section 2, or at least before going into section 3) for incorporating distributions of the return/risk in the offline RL setting, along with visualizations of what the proposed method CODAC is designed to do (e.g. manage risk while learning offline).

quality: the theoretical results are intuitive, appealing and easy to understand, particularly in the context of existing algorithms in offline RL and distributional RL. the experimental evaluation is quite thorough, testing the ability of CODAC to learn both risk-neutral and risk-sensitive policies, on data-sets of various quality, and baselines appear to be well chosen and comprehensive (see limitations for comments).

significance: offline reinforcement learning and risk-sensitive distributional RL are both very active and emerging topics in reinforcement learning. based on this observation, I believe this paper will have a very high impact in the research community.

**Time Spent Reviewing:**

1.5

---

> ### Author Response · Authors · 2021-08-10
> **Response to Reviewer 8SBh**
>
> We thank the reviewer for their thoughtful feedback. We will incorporate the writing suggestions in our next draft. Below, we answer the reviewer's questions and provide additional experimental results. Please let us know if we can provide any additional clarifications during the discussion period.
>
> ---
> **Comment 1**: Are CODAC quantile lower bounds reasonably tight?
>
> **Response 1**: Yes, our lower bounds are tight up to the quantile function concentration bound $\Delta(s,a)$, analogous to the CQL Q-value lower bound dependency on the transition function concentration bound. Here, we provide a proof sketch of this result. Following a very similar argument to Equation 8 and 9 in Appendix, by Lemma A.1, we have that with probability at least $1-\delta$,
>
> \begin{equation}
>  \quad F^{-1}_{\tilde{\mathcal{T}}^\pi Z^\pi (s,a)}(\tau)
> \end{equation}
>
> \begin{equation}
> = F^{-1}_{\hat{\mathcal{T}}^\pi Z^\pi (s,a)} (\tau) - c(s,a)
> \end{equation}
>
> \begin{equation}
> \geq F^{-1}_{\mathcal{T}^\pi Z^\pi(s,a)} (\tau)- c(s,a) - \Delta(s,a)
> \end{equation}
>
> Then, by re-arranging and Lemma A.6, we have
>
> \begin{equation}
> \quad F^{-1}_{Z^\pi(s,a)}(\tau)
> \end{equation}
>
> \begin{equation}
> \leq F^{-1}_{\tilde{\mathcal{T}}^\pi Z^\pi (s,a)}(\tau) + c(s,a) + \Delta(s,a)
> \end{equation}
>
> $\leq F^{-1}_{\tilde{\mathcal{T}}^\pi Z^\pi (s,a)}(\tau)$
>
> $ \quad + \max_{s',a'} c(s',a') + \Delta(s',a') $
>
> $\leq F^{-1}_{\tilde{Z}^\pi (s,a)}(\tau)$
>
> $ \quad + (1-\gamma)^{-1} \max_{s',a'}\{c(s',a') + \Delta(s',a')\}$
>
> Thus, we see that the lower bound is tight up to the polynomial factors that appear in $\Delta(s,a)$ (see Lemma A.1). This matches our intuition that as the number of samples for every state-action pair $n(s,a)$ increases, the bound will also become tighter (because $\Delta(s,a)$ decreases). Note that the bound depends on $c(s,a)$ (which depends on our choice of $c_0(s,a)$), but as we show at the end of Appendix A.2 (after Equation 9), it suffices to take $c_0(s,a)$ in a way that $c(s,a)$ is linear in $\Delta(s,a)$. We will include this derivation as well as a discussion about the tightness of our quantile lower bound in our next paper draft.
>
> ---
> **Comment 2**: Interpretation of stochastic D4RL results.
>
> **Response 2**: To better interpret the stochastic D4RL results, we have collected behavioral statistics of the agents trained on the risk-sensitive HalfCheetah-Mixed-v0 and Walker2d-Mixed-v0 datasets. In these environments, a stochastic penalty is applied to the reward at each timestep if the cheetah/walker's horizontal/angular velocity exceeds the threshold of $4/0.5$ (See Appendix C.2 for details). We execute one trained agent for each method reported in Table 2 for 10 episodes in the environment and record the percentage of timesteps where the agent violates the threshold and their average velocity over these evaluation episodes.
>
> As shown in the table below, CODAC-C achieves the lowest percentage of violations in the HalfCheetah environment, indicating that it has learned a safer policy than all other methods. On Walker2d, CQL appears to be the safest; however, this is due to the fact that CQL failed to learn the desirable walking behavior as indicated by its low reward in the paper. Among the methods that learned to walk, CODAC-C achieves the lowest average angular velocity while maximizing the return.
>
> |                | HalfCheetah-Mixed-v0 |                  | Walker2d-Mixed-v0 |                  |
> |:--------------:|:--------------------:|:----------------:|:-----------------:|:----------------:|
> |    Algorithm   |      % Violation     | Average Velocity |    % Violation    | Average Velocity |
> | CODAC-C (Ours) |        **11**        |     **1.49**     |         15        |     **0.28**     |
> | CODAC-N (Ours) |          54          |       2.02       |       **9**       |       0.34       |
> |       CQL      |          23          |       1.71       |         13        |       0.19       |
> |      ORAAC     |          37          |       1.76       |         48        |       0.49       |
>
> ---
> **Comment 3**: Can CODAC be applied to other risk measures such as Wang?
>
> **Response 3**: Yes, we thank the reviewer for pointing out this connection. Many common alternative risk measures such as Wang and Cumulative-Probability-Weighting (CPW) are also distorted expectations of the quantile values, and our lower bound results readily extend to these objectives. As such, our theoretical contribution extends to an unified framework for optimizing any quantile-based policy objective in the offline setting.
>
> In the current submission, we focused on CVaR for its wide adoption in risk-sensitive RL, but we will include a discussion and experiments on these alternative risk objectives in our next update and highlight our approach's generality. Here, we present some preliminary results on applying CODAC to these alternative risk objectives. In particular, we train CODAC using the Wang (CODAC-Wang) and CPW (CODAC-CPW) objectives on the Risky PointMass domain and report the results below; CODAC-CVaR and CODAC-Neutral results are taken directly from the paper.
>
> |               | Risky PointMass |           |            |            |
> |:-------------:|:---------------:|:---------:|:----------:|:----------:|
> |   Algorithm   |       Mean      |   Median  |  CVaR-0.1  | Violations |
> |   CODAC-Wang  |    **-6.01**    |   -4.46   |   -16.80   |     7.0    |
> |   CODAC-CVaR  |      -6.05      |   -4.89   | **-14.73** |   **0.0**  |
> |   CODAC-CPW   |      -8.34      | **-4.00** |   -54.18   |    103.0   |
> | CODAC-Neutral |      -8.60      |   -4.05   |   -51.96   |    108.3   |
>
> As can be seen, CODAC-W performs similarly to CODAC-CVaR (CODAC-C in the paper), trading off slightly better average performance at the cost of safety. On the other hand, CODAC-CPW is on par with CODAC-Neutral (the risk-neutral version). These findings match with our intuition that Wang is slightly more risk-seeking than CVaR since it gives non-zero (but vanishingly small) weight to quantile values above the risk cutoff threshold, and CPW is similar to risk-neutral due to its intended modeling of human game-play behavior. These findings are also consistent with those in prior work [A], which investigates these risk objectives for online distributional reinforcement learning.
>
> Finally, to the best of our knowledge, we do not believe that our framework extends to exponential utilities or certainty-equivalents, as they do not admit expectations over return quantiles. For our next paper draft, we will expand these results in detail.
>
> [A] Dabney, Will, et al. "Implicit quantile networks for distributional reinforcement learning." International conference on machine learning. PMLR, 2018.
>
> ---
> **Comment 4**: Why does CODAC underperform sometimes (e.g., risk-sensitive HalfCheetah) and what can be done in the future to address this?
>
> **Response 4**: We found that when CODAC underperforms, the training curve often makes substantial progress but experiences extended periods of performance degradation that leads to the eventual sub-optimal solution. Additionally, as stated in the conclusion of the main text, offline RL algorithms, including CODAC, are sensitive to hyperparameters. Therefore, we believe the most effective way for improving CODAC would be developing  early-stopping methods as well as hyperparameter selection strategies for offline RL algorithms, which remain open problems in the field. We intend to investigate these research directions in future work.
>
> Specific to CODAC, we have found one way to improve its training stability is to decrease the frequency of penalized quantile re-sampling (see Line 14 in Algorithm 1 in Appendix). Right now, we sample a new quantile to apply CODAC penalty in each critic update, and we hypothesize that this introduces a highly non-stationary critic for the actor update, which is particularly problematic in the offline setting due to the lack of new online data collection; thus, any bias will propagate without correction, leading to unstable training.
>
> To address this issue, we consider a variant of CODAC, in which the penalized quantile is re-sampled every 5 updates instead. Using this variant, we have retrained CODAC-C on the risk-sensitive HalfCheetah environment using the same hyperparameters as in the text on 3 random seeds. We obtained average return of $338.47$ and CVaR of $90.29$, which greatly improves upon the reported results in the paper and matches the performance of the best performing algorithm, ORAAC. We intend to investigate further by training this CODAC variant on other datasets and report the results in the Appendix section of our next updated paper.

---

> > ### Comment · Reviewer_8SBh · 2021-09-01
> > **Thanks for addressing my comments**
> >
> > Thank you for your detailed response. I have read your response to my questions and those of the other reviewers and happy to keep my score. I would recommend to include the tightness proof and explain its intuition in the main paper, as well as the updated results you provided above. I also do believe that illustrating the proposed approach with a conceptual diagram/workflow, and how it differs from conventional CQL could be very helpful here.

---

### Author Response · Authors · 2021-08-10
**Response to All Reviewers**

Dear reviewers,

Thank you for your constructive and thoughtful feedback. In our responses to each individual reviewer, we have obtained new theoretical and experimental results that may be of interest to other reviewers as well. Here, we aggregate and highlight the new results for the convenience of all reviewers. In particular,
- We have shown that the CODAC quantile lower bound is tight (Response 1, Reviewer 1).
- We have shown that CODAC can be applied to alternative quantile-based risk-sensitive objectives, such as Wang and CPW (Response 3, Reviewer 1).
- We have shown that CODAC is compatible with augmented state space for optimizing CVaR (Response 2, Reviewer 3).
- We have included qualitative analysis of the stochastic D4RL results (Response 2, Reviewer 1).

Below, we have provided responses to each individual reviewer’s comments. Please let us know if we can provide any additional clarifications during the discussion period.

Best,

Authors

---

### Decision · Program_Chairs · 2021-09-27

**Decision:**

Accept (Poster)

**Comment:**

All reviewers agree that this is a good paper with sufficient novelty and relevance to the NeurIPS community. Reviewers had several questions, which were adequately answered by the authors. Therefore, I would recommend the *acceptance* of this paper.

I encourage the authors to consider the feedback by reviewers in their revision of the paper, including the followings:

- discuss the tightness of the lower bound
- discuss what other risk measures CODAC can be applied to
- discuss that CVaR's not being time-consistent, and how you avoid it.
- cite and compare with the literature on moment matching for DRL
- Fix the typos, including the one in Theorem 3.6